# ScImage: How Good are Multimodal Large Language Models at Scientific Text-to-Image Generation?

**Leixin Zhang**[1], **Steffen Eger**[2], **Yinjie Cheng**[3], **Weihe Zhai**[4], **Jonas Belouadi**[5], **Fahimeh Moafian**[6], **Zhixue Zhao**[3]

[1]University of Twente `l.zhang-5@utwente.nl`
[2]University of Technology Nuremberg `steffen.eger@utn.de`
[3]University of Sheffield `{ycheng80,zhixue.zhao}@sheffield.ac.uk`
[4]Harbin Institute of Technology `weihezhai@insun.hit.edu.cn`
[5]University of Mannheim `jonas.belouadi@uni-mannheim.de`
[6]Technische Universität Dresden `fahimemoafian@gmail.com`

## Abstract

Multimodal large language models (LLMs) have demonstrated impressive capabilities in generating high-quality images from textual instructions. However, their performance in generating scientific images—a critical application for accelerating scientific progress—remains underexplored. In this work, we address this gap by introducing ScImage, a benchmark designed to evaluate the multimodal capabilities of LLMs in generating scientific images from textual descriptions. ScImage assesses three key dimensions of understanding: spatial, numeric, and attribute comprehension, as well as their combinations, focusing on the relationships between scientific objects (e.g., squares, circles). We evaluate seven models, GPT-4o, Llama, AutomaTikZ, Dall-E, StableDiffusion, GPT-o1 and Qwen2.5-Coder-Instruct using two modes of output generation: code-based outputs (Python, TikZ) and direct raster image generation. Additionally, we examine four different input languages: English, German, Farsi, and Chinese. Our evaluation, conducted with 11 scientists across three criteria (correctness, relevance, and scientific accuracy), reveals that while GPT-4o produces outputs of decent quality for simpler prompts involving individual dimensions such as spatial, numeric, or attribute understanding in isolation, all models face challenges in this task, especially for more complex prompts.[1]

## 1 Introduction

Artificial intelligence (AI) has become an increasingly valuable tool in academic research, offering support across various aspects of the scientific process (Byun & Stuhlmüller, 2023; Chen & Eger, 2023; Lu et al., 2024a; Nechakhin et al., 2024; Shao et al., 2024). For instance, AI-based tools, such as Deep Research [2], Elicit (Byun & Stuhlmüller, 2023)[3] and ResearchRabbit[4], facilitate finding relevant literature for specific research topics. Tools like Grammarly assist with grammatical refinement and phraseology in academic writing and LLM-assisted text production is nowadays common (Liang et al., 2024). LLMs can also generate new ideas for scientific papers that rival the ideas produced by human scientists (Si et al., 2024). Even more holistically, approaches like The AI Scientist (Lu et al., 2024a) have demonstrated the capability to generate entire research output, encompassing everything from initial conceptualization to experimental design and paper drafting.

Despite these advancements, a critical subproblem remains relatively unexplored: the AI-driven generation of scientific visualizations, including illustrative figures, charts, and plots (Voigt et al., 2024). These visual elements play a pivotal role in scientific communication (Lee et al., 2016), serving as essential tools for researchers, educators, and students to convey complex ideas, data, and concepts. The ability to automate the creation of accurate scientific images from textual descriptions could significantly enhance both the efficiency and effectiveness of scientific communication and production. Compared to previous attempts at automating image generation, AI-driven generation of scientific visualizations does not strictly rely on tabular data input (Yamada et al., 2018), does not require cumbersome parameter adjustments (Lindsay et al., 2017), and produces a variety of outputs beyond just statistical images (Waskom, 2021).

---

[1]ScImage prompts: `https://huggingface.co/datasets/casszhao/ScImage` Code: `https://github.com/leixin-zhang/Scimage`

[2]https://openai.com/index/introducing-deep-research/

[3]https://elicit.com/

[4]https://www.researchrabbit.ai/

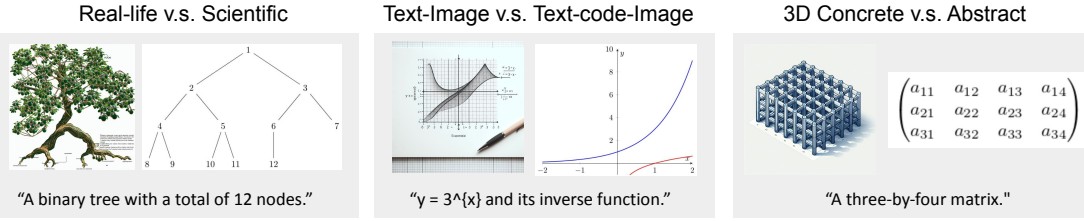

Figure 1: Illustration of scientific text-to-image generation. The text shown below is the generation query. Images on the left meet the expectations for general text-to-image tasks, while those on the right highlight the specific requirements of scientific image generation. All figures are from our `ScImage` experiments.

Scientific visualizations often require precise spatial composition, accurate numeric representations, and correct attribution of complex scientific objects. These elements must be combined in ways that adhere to established conventions within scientific domains. While general-purpose text-to-image models have made significant strides (Esser et al., 2024; Touvron et al., 2023; Ramesh et al., 2021), the requirements, e.g., precise and high-resolution graphical representations, pose unique challenges for scientific image generation, as illustrated in Figure 1. Moreover, the representation of objects in scientific domains—such as batteries in circuit diagrams or trees in graph theory—differs significantly from their appearance in real-life images.

In response to this need, we present `ScImage`, a comprehensive benchmark aimed at evaluating the capabilities of multimodal LLMs in generating scientific images conditioned on textual descriptions. Our benchmark includes a diverse set of skills that test key dimensions of scientific image production individually and in combination, covering a wide range of scientific objects, their attributes, and relations. As scientific figures are often generated with high-level coding languages such as TikZ or Python, we evaluate standard LLMs (all capable of generating code output) such as LLAMA 3.1 8B and GPT-4O, in addition to inherent multimodal models such as DALL·E on `ScImage`.

Our findings highlight the need for continued research in enhancing the capabilities of multimodal LLMs for scientific image generation. By providing a standardized benchmark and in-depth analysis, `ScImage` aims to drive progress in this critical area, ultimately supporting more efficient scientific image production.

Key contributions of this work include:

- We provide a benchmark, `ScImage`, for testing the model capability of scientific text-to-image generation along (predominantly) three understanding dimensions: numeric, spatial, and attribute comprehension.
- We explore seven state-of-the-art models on `ScImage`, including code-based and genuine multimodal.
- We comprehensively assess the models using almost a dozen human scientists[5] across three evaluation aspects: correctness, relevance, and scientificness, and four languages: English, German, Chinese, and Farsi.
- We analyze model performances across different object types, comprehension dimensions, and input languages.
- We provide human evaluation scores for ~3k generated scientific images, totaling an annotation value of approximately 3,000 USD.[6] These evaluation scores serve as a "ground truth" for the evaluation of generation performance and support future research on developing automated metrics for assessing scientific images.

## 2 RELATED WORK

In computer vision and multimodal studies, there are many benchmarks and datasets serving various purposes, including object detection (Lin et al., 2014), image classification (Krizhevsky, 2009, Deng et al., 2009), hand-written digits recognition (Deng, 2012), and image captioning (Sharma et al., 2018, Chen et al., 2015), but the majority focus on real world images. Although datasets like Paper2Fig (Rodriguez et al., 2023) and DaTikZ (Belouadi et al., 2024a;b) include scientific figures and captions extracted from research papers, there is no structured evaluation of the limitations

---

[5]Our scientists are PhD students and higher. We use the term 'scientist' to differentiate them from crowd-workers or early career academics such as Bachelor students.

[6]For English, we evaluate 404 prompts for 7 different models, yielding 2828 individual images (308 of which have compile errors, receiving an automatic score of zero). For the later multilingual phase, we evaluate 540 images (58 with compile errors). In total, we thus evaluate 3368 generations, 366 of which have compile errors.

and capabilities of scientific text-to-image models. In Section 2.1 and 2.2, we review existing benchmarks designed to assess model abilities: visual understanding (e.g., using images as inputs (Thrush et al., 2022; Huang et al., 2023; Wu et al., 2024), discussed in Section 2.1) and ability of text-to-image generation (Section 2.2). All surveyed datasets and benchmarks in this study are summarized in Table 7 in Appendix A.

## 2.1 IMAGE AS INPUT

Benchmarks that use images as input often take the form of visual question answering (VQA), where images are paired with questions about their content (Biten et al., 2019; Das et al., 2024; Yue et al., 2023; Wang et al., 2024a). For example, the Multimodal Visual Patterns (MMVP) Benchmark (Tong et al., 2024) focuses on challenging cases, comprising 150 CLIP-blind pairs (images that the CLIP model perceives as similar despite clear visual distinctions) with questions designed to probe specific image details, such as relative position, object counting, or other attributes.

In the scientific domain, VQA examples are typically sourced from exams, quizzes, or textbooks (Yue et al., 2023; Lu et al., 2024b; Li et al., 2024a). Additionally, ScienceQA (Lu et al., 2022) is a benchmark that uses images as contextual inputs for questions, rather than directly asking about the image's content. This dataset also incorporates Chain of Thought (CoT) reasoning to enhance interpretability alongside the answers. CharXiv (Wang et al., 2024b) evaluates models' abilities to describe and reason about charts through multiple-choice questions.

Another type of visual understanding benchmark focuses on caption-image alignment. Winoground (Thrush et al., 2022), for instance, challenges models to match images with their corresponding captions. The dataset includes pairs where objects or predicates are swapped, such as "there is a mug in some grass" versus "there is some grass in a mug", to test fine-grained comprehension of texts. MMSCI (Li et al., 2024b) extends this focus to the scientific domain, offering a figure-captioning benchmark spanning 72 subjects. Additionally, SciFIBench (Roberts et al., 2024) evaluates figure-caption alignment through tasks such as selecting the appropriate figure for a given caption or choosing the correct caption for a specific figure from multiple choices.

## 2.2 IMAGE AS OUTPUT

Compared to visual understanding, benchmarks that assess individual dimensions of abilities in text-to-image generation models remain relatively scarce. One benchmark designed for this purpose is T2I-CompBench (Huang et al., 2023), which includes 6k compositional text prompts, categorized into three groups: attribute binding (e.g., color and shape), object relationships (e.g., spatial arrangements), and complex compositions. While we are inspired by this benchmark, we note that it does not target the scientific domain.

In the context of vector graph and scientific figure generation, Zou et al. (2024) develop an evaluation set to assess models' abilities in prompt comprehension and vector graph generation. Belouadi et al. (2024a) introduce a dataset that pairs scientific paper captions (as input) with TikZ code (as output), which can be compiled into vector graphs. Additionally, Shi et al. (2024) explore models' capabilities to replicate chart images by converting them into Python code. Compared to these works, which focus on specific evaluation settings (such as TikZ or vector graphic or chart generation), our evaluation setup is broader, more targeted and more structured: we assess model performance across different input languages and output formats (TikZ vs. Python vs. plain image), object types and aspects of understanding.

## 2.3 EVALUATION OF TEXT-TO-IMAGE MODELS

Existing evaluations of text-to-image models primarily focus on text-image alignment and image quality for real-world images, as demonstrated by benchmarks such as MS COCO (Lin et al., 2014) and studies in Sharma et al. (2018) and Chen et al. (2015). Later works, such as Lee et al. (2024) and Cho et al. (2023), broaden the scope of evaluation to include aspects like aesthetics, originality, social bias, and efficiency, but these still remain within the domain of real-world images.

Benchmarks designed for evaluating real-life images are insufficient for assessing the quality of generated scientific graphs. Compared to general images, scientific graphs must prioritize accuracy in representing scientific concepts and ideas. This includes ensuring the precision of numerical values in charts or plots, and adhering to established conventions when translating real-world objects into graphical representations (e.g., depicting a "battery" in electric circuit diagrams within the domains of engineering and physics).

For figures in the scientific domain, metrics like CLIPScore and Fréchet Inception Distance (FID) are employed to assess the quality of generated graphics (Zou et al., 2024). Additionally, Shi et al. (2024) utilize GPT-4V for automated evaluations of image quality. However, as highlighted by the MMVP benchmark (Tong et al., 2024), automated

evaluations can be unreliable, particularly when recognizing precise directions in text and images, such as "up" and "down". The precision required for evaluating scientific graphs presents significant challenges for current automated metrics. To address this gap, we resort to human evaluation instead of automatic evaluation, using a panel of 11 scientists. Additionally, we show that, indeed, standard multimodal metrics employed in the community have low correlations to our human annotators in our scientific domain.

# 3  SCIMAGE

## 3.1  TASK SETUP

ScImage evaluates the capability of multimodal LLMs to generate scientific graphs from textual descriptions. We design prompts that require models to understand and visualize scientific concepts, emphasizing three key dimensions of understanding: (a) **Spatial understanding**: Assessing the models' ability to interpret and represent spatial relationships between objects, such as "left of" and "on top of". (b) **Numeric understanding**: Evaluating the models' capacity to handle and visualize numerical requests accurately, such as the exact number of objects or requests like 'more' and 'half'.[7] (c) **Attribute binding**: Testing the models' ability to correctly represent object attributes such as color, size, and shape. Figure 2 demonstrates these three key dimensions of understanding.

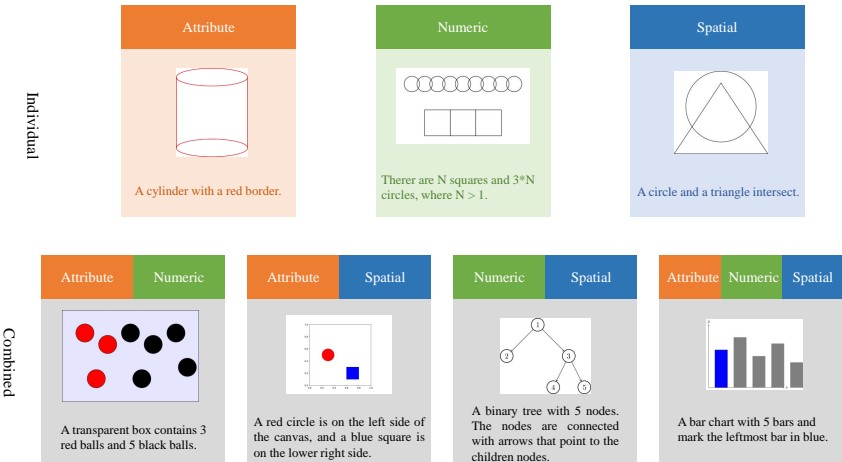

Figure 2: Illustration of the three understanding dimensions. The first row shows the individual dimensions of Attribute, Numeric and Spatial understanding. The second row illustrates the combination of two or three dimensions.

**Output mode** The task involves generating images either (i) *directly text-image* or (ii) *text-code-image* through intermediate code (Python or TikZ) — which then has to be compiled to images — based on textual prompts.

**Prompting** We instruct the models to generate scientific graphs with prompts. Each prompt consists of an *auxiliary instruction* and a *generation query*. The auxiliary instruction is used to constrain the model to generate scientific graphs in either (i) direct text-image or (ii) text-code-image mode. Language models can exhibit sensitivity to variations in prompts (Leiter & Eger, 2024). To mitigate the impact of this variability and ensure a fair comparison between models, we conduct pilot tests to find prompts that generally lead all tested models to generate required output type (i.e., Python code, TikZ code, or images) in a scientific style. Our resulting auxiliary instructions are shown in Table 8 in Appendix B. Examples of code and image output are presented in Appendix D and Appendix E.

## 3.2  DATASET CONSTRUCTION

We begin with a comprehensive survey of relevant scientific datasets and benchmarks, as detailed in Table 7 in Appendix A, also including math and science textbooks. This gave us the intuition that scientific graphs are described by objects and their properties (attributes) as well as their relative positioning (spatial relations) and numeric information (e.g., how many objects). Additionally, annotations often emphasize parts of the scientific image.

---

[7]The articles 'a' and 'an' are not interpreted as numerical descriptors.

Thus, we develop prompt requirements to ensure that varying aspects of scientific text generation are covered. We require that each prompt must explicitly define: (a) the core visual elements (objects) to be generated in the graph, e.g. cycle, square, etc.; (b) specific attributes of the object (**attribute binding**), e.g., red cycle, or count of the object (**numeric**), e.g. three cycles. the positioning arrangement and placement (**spatial**) of objects within the graph, e.g. on the bottom or in relation to another object (to the left). (d) We finally consider any required labels, legends, or additional textual elements (annotations). Further, for graphs containing multiple objects, the prompt must additionally specify the quantity of each object type, the relative spatial or logical relationships between objects, and the individual properties of each object group. Individual aspects are typically optional, i.e., not every prompt has to specify numerical or spatial components. Specific details on dataset construction follow below.

**Generation Queries** $Q$    We adopt a structured methodology that leverages a **dictionary** $D$ along with a set of **query templates** $T$ to create a diverse, comprehensive, and traceable set of generation queries $Q$ for the `ScImage` evaluation dataset.

**Dictionary** $D$ defines key elements relevant to scientific figures, including objects (e.g., square and circle), attributes (e.g., color and size), spatial relations (e.g., left, right, between), and numeric values (e.g., three, five, two more). To construct this dictionary, we begin by building the list of objects. First, we manually extract frequent object entities from DATIKZ (Belouadi et al., 2024a), an image-caption dataset derived from scientific publications.

We filter out objects highly dependent on the context of the original paper, such as mathematical formulas adjacent to figures and line segments with specific values.[8] Additionally, we simplify complex objects—such as intricate circuit designs and automata intended for specific applications. Next, we manually define representative spatial relations, such as "to the left" and "at the center of", to describe the positions of objects within graphs. Additionally, we create attribute sets to capture detailed object properties, including size, color, and line thickness. Numerical requests are also incorporated into the dictionary to replace values in bar charts or assess models' accuracy in representing object counts. We then collect attributes and relations that appear at least three times and then manually compile a list of attributes and spatial relations by merging similar ones and removing domain-specific ones.

At the top level, $D$ is organized into classes for objects, attributes, numeric, and spatial relationships. Each class then contains a list of descriptive words specifying the class. When selecting a descriptive word from $D$ for a given blank in the query templates $t_i$, we first locate the specific list corresponding to the word class and then randomly choose an item from it.[9] For clarity, we present a snippet of $D$ below:

```
D = {"2D_objects": ["square", "circle", ...],
     "3D_objects": ["cube", "sphere", ...],
     "colors": ["red", "blue", ...],
     "spatial_relations": ["left", "right", ...], ...}
```

We define a set of **query templates** $T$, where each template $t_i \in T$ is one or several sentences with one or more placeholders. These placeholders are designed to be filled with elements $d_j \in D$, which may include objects, attributes, or relations. To construct a generation query $q_k \in Q$, we select a query template $t_i$ and populate its placeholders with appropriate attributions from $D$. This process allows us to create diverse queries with meta-info. The final step in preparing our generation prompt involves prefixing the constructed generation query $q_k$ with task-specific auxiliary instructions, as detailed in Section 3.1. This structured approach ensures transparency and flexibility across diverse queries, while minimizing potential biases toward specific objects, attributes, and other contextual elements.

Further, `ScImage` is crafted to cover a wide range of scientific graph types and complexities. We classify the object types of all prompts based on the following categories: 2D geometric shapes, 3D geometric shapes, charts, graph theory representations, matrices, real-life object modeling, tables, additional annotations, functions & coordinates. For each category, we create multiple templates that vary in complexity and combine different aspects of spatial, numeric, and attribute understanding. For each template, we create four different generation queries by sampling different elements from the dictionary $D$. In total, we have 101 query templates and 404 generation queries. Examples of `ScImage` are shown in Table 9.[10]

---

[8]DATIKZ is unsuitable for our exploration, as it contains textual descriptions 'in the wild' and additionally its captions are often unsuitable for reconstructing the output image at hand.

[9]During the word collection process, we excluded complex and rare terminologies due to potential bias. Consequently, attributes such as 'hinges' and 'hyperstatic structures' were omitted.

[10]In our dataset construction, we aimed to provide clearly specified target dimensions for scientific image generation, where the queried attributes are precisely known. This approach enables specific and targeted evaluation. It contrasts with 'real-world' evaluations that often utilize lengthy query prompts encompassing multiple dimensions simultaneously, where model failures cannot be clearly disentangled across different aspects. For instance, while Datikz (Belouadi et al., 2024) provided a real-world dataset that

## 3.3 EVALUATION

We employ a multi-faceted evaluation approach to assess the quality and accuracy of the generated scientific graphs:

| Agreement | Correctness | Relevance | Scientificness |
|---|---|---|---|
| Joint Spearman $r$ | 0.67 | 0.62 | 0.73 |
| Joint Pearson $r$ | 0.70 | 0.64 | 0.71 |
| Joint Weighted Kappa | 0.50 | 0.41 | 0.45 |
| Pair Spearman $r$ (Eng) | 0.73 | 0.64 | 0.63 |
| Pair Pearson $r$ (Eng) | 0.75 | 0.65 | 0.63 |
| Pair Weighted Kappa (Eng) | 0.61 | 0.52 | 0.47 |
| Pair Spearman $r$ (Multi) | 0.80 | 0.75 | 0.73 |
| Pair Pearson $r$ (Multi) | 0.80 | 0.77 | 0.79 |
| Pair Weighted Kappa (Multi) | 0.64 | 0.60 | 0.66 |

Table 1: Agreement of joint evaluation in the calibration session (joint) and pair agreement (agreement across all examples of two sets of annotations) in the final evaluation for English and multilingually.

**Human Evaluation** Our human evaluators assess the generated images based on three criteria: **Correctness**: Assessing the accuracy of the visual representation in relation to the textual prompt. **Relevance**: Evaluating how well the model avoids generating redundant or irrelevant objects or attributes of objects. **Scientific Style**: Evaluating the appropriateness of the image for use in scientific publications. Each criterion is rated on a scale from 1 to 5, with 5 being the highest score. An additional score of 0 is assigned in cases where code generation cannot be compiled into an image due to compilation errors.[11] The detailed evaluation guideline is given in Appendix F.

We employ a panel of 11 expert annotators, carefully selected to represent the target users of scientific plots and graphs. This panel consists of: eight Ph.D. students, one postdoctoral researcher, and two faculty members from mathematics and computer science, ensuring domain expertise in evaluating scientific visualizations. We provide each annotator with detailed annotation guidelines given in the Appendix F. Furthermore, before formal annotation distribution, we conduct a **calibration session** to match the understanding of the annotation standard. We randomly assign examples to annotators and further assign at least two annotators per instance to mitigate annotation biases.

**Agreements** Table 1 presents the agreement scores (Spearman's $r$, Pearson's $r$ and weighted Kappa) from both the small-scale calibration session (joint evaluation: 315 images are evaluated by all evaluators) and the later pairwise evaluation (pair evaluation: every image is evaluated by a pair of evaluators) using various models (see Section 4). A relatively strong positive correlation with Pearson $r$ and Spearman $r$ between 0.62 and 0.80 is observed across all evaluation criteria. Weighted kappa, a chance-corrected measure of agreement for ordinally scaled samples, is within commonly accepted ranges for agreement, with almost all measures above 0.5. The multilingual evaluation, conducted by a subset of 6 more experienced evaluators, shows a higher level of agreement than the English evaluation. Overall, weighted kappa is 0.62 for correctness for combined English and multilingual evaluation and above 0.52 for relevance and scientificness.

We tax the value of our evaluation at roughly 3,000 USD, with up to 11 annotators involved for up to 7 hours each, all working for a conservative estimate of 40 USD per hour (including taxes), on average.

**Automatic Evaluation** We also test how well recent automatic text-to-image evaluation metrics correlate with our human judgements. We explore 5 recent multimodal metrics. These achieve a highest Kendall correlation with human scores of 0.26, where the agreement on the correctness dimension is highest and lowest on scientificness (maximum Kendall of 0.15). This underscores the value and necessity of our human annotation campaign. Details are given in Appendix H.

## 4 EXPERIMENTS

We employ two types of models for image generation, corresponding to the two output modes described in Section 3.1. For (i) direct text-image mode, we include DALL·E and STABLE DIFFUSION; for (ii) text-code-image, we include GPT-4o, LLAMA 3.1 8B and AUTOMATIKZ (Belouadi et al., 2024a), where the model is prompted to generate Python or TikZ code. AUTOMATIKZ is specifically fine-tuned for generating TikZ code, therefore, we only prompt AUTOMATIKZ to generate TikZ.

**Overall performance**: The overall model results are presented in Table 2, showing averages based on per-instance human evaluations and further averaged across annotator pairs. Instances where code generation resulted in compilation

---

inspired our selection of scientific objects, it lacks specificity and combines various understanding properties within each prompt. Our evaluation framework draws inspiration from the established CheckList methodology of Ribeiro Ribeiro et al. (2020).

[11]While assigning a score of 0 may seem harsh, we also report results when ignoring all compile errors, which would constitute an upper bound.

errors are penalized with a score of 0.[12] In Appendix G, we show scores where compilation errors are ignored; potentially compilation errors could be fixed by self-consistency checks in future approaches. Overall, *GPT-4o in text-code-image mode (GPT-4o_tikz and GPT-4o_python) stands out as the model achieving the best scores across all evaluation dimensions. However, it scores below 4 in all three evaluation dimensions, indicating at least some mistakes on average in every output.*

| Model | Correctness | Relevance | Scientific style | Compile Error Rate |
|---|---|---|---|---|
| Automatikz | 2.05 | 2.31 | 3.35 | 0.04 |
| Llama_tikz | 1.78 | 1.94 | 2.61 | 0.29 |
| GPT-4o_tikz | 3.50 | **3.67** | 3.75 | 0.09 |
| Llama_python | 2.10 | 2.54 | 3.18 | 0.28 |
| GPT-4o_python | **3.51** | 3.40 | **3.93** | 0.07 |
| Stable Diffusion | 2.19 | 2.09 | 1.96 | - |
| DALL·E | 2.16 | 2.00 | 1.55 | - |

Table 2: Overall model performance, averaged across instances and annotators (compile error of code output is penalized with score 0).

**Correctness**: GPT-4O outperforms all other models by a large margin (more than 1.3 points), achieving the highest scores for both the text-tikz-image (3.50) and text-python-image (3.51) output. All other models have correctness scores between 1.7 and 2.2, indicating low correspondence between instruction and output image. LLAMA 3.1 8B_tikz (1.78) and AUTOMATIKZ (2.05) are the worst models. Interestingly, LLAMA 3.1 8B performs considerably better with Python as output than with TikZ as output (2.10 vs. 1.78). Similarly, GPT-4O performs marginally better with Python as the coding language. When ignoring compile errors (Table 19), the code generation models become substantially better, especially LLAMA 3.1 8B improves by almost 1 point, leaving the direct text-image generation models STABLE DIFFUSION and DALL·E among the worst models.

**Relevance**: GPT-4O also dominates relevance, but with a smaller margin (1.13 vis-a-vis the second best model LLAMA 3.1 8B with Python output). LLAMA 3.1 8B again prefers Python output while GPT-4O prefers TikZ. Visual models STABLE DIFFUSION and DALL·E often include irrelevant details in their output and are among the worst; see Table 10 in Appendix E for examples.

**Scientificness**: Images converted from Python or TikZ code receive notably higher scores ($> 2.5$) in scientific style compared to direct image generation from DALL·E and STABLE DIFFUSION ($< 2$). This suggests that code output as an intermediate step offers a significant advantage for scientific graph generation, as opposed to visual models that are primarily trained on real-life images.

However, a significant drawback of models that generate code is the potential for compilation failures. For instance, GPT-4O experiences 35 TikZ code and 27 Python code compilation errors. LLAMA 3.1 8B has even much higher failure cases: 116 for TikZ mode and 113 for Python code, representing approximately 28% of all prompts. `Automatikz` performs best in terms of compilation success, with only 17 cases in total. The low scores observed for LLAMA 3.1 8B in Table 2 can largely be attributed to penalties for these compilation errors. If these were ignored (or could be fixed), LLAMA 3.1 8B would outperform AUTOMATIKZ as shown in Table 19 in Appendix G (note, however, that the comparison for both models includes different instances, thus is not fully fair in the table).

## 5 ANALYSIS

We conduct a more detailed analysis of model performance, focusing on understanding types (attribute, numerical and spatial understanding) and object types to identify which categories present the greatest challenges for the models.

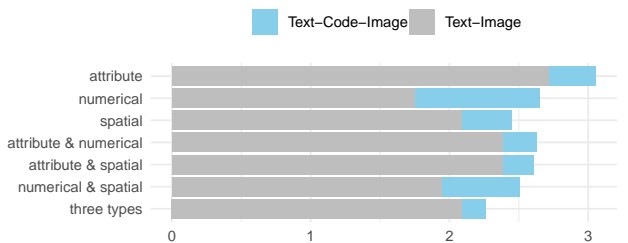

Figure 3: Comparison of text-code-image and text-image: correctness scores averaged across model types. 'Three types' means attribute, numerical and spatial understanding combined.

**Types of Understanding**  Table 3 presents the fine-grained correctness scores for different understanding types. Figure 3 illustrates the performance of two modes of generation (text-code-image and text-image) separately. Notably, spatial understanding appears to be the most challenging across all textual models. For instance, while GPT-4O achieves scores around 4.0 for attribute binding, its performance drops substantially for spatial understanding, remaining well below 3.5.

---

[12]A score of 0 does not apply if a compile error occurs due to missing TikZ code like $\backslash begin\{documentclass\}$. Models sometimes assume that a document has already been set up, so we check and add codes at head and tail for generated codes before compiling the image.

| Model | Attribute | Numerical | Spatial | Attribute & Numerical | Attribute & Spatial | Numerical & Spatial | Attribute & Numerical & Spatial |
|---|---|---|---|---|---|---|---|
| Automatikz | 2.42 | 1.91 | 1.71 | 2.29 | 2.04 | 2.13 | 1.77 |
| Llama_tikz | 2.53 | 1.55 | 1.77 | 1.69 | 1.84 | 1.91 | 1.3 |
| GPT-4o_tikz | 4.11 | 3.49 | 3.35 | 3.41 | 3.53 | 3.59 | 3.13 |
| Llama_python | 2.24 | 2.38 | 1.96 | 2.28 | 2.28 | 1.63 | 1.97 |
| GPT-4o_python | 3.95 | 3.92 | 3.47 | 3.46 | 3.34 | 3.28 | 3.13 |
| Stable Diffusion | 2.75 | 1.73 | 2.06 | 2.41 | 2.46 | 1.96 | 2.11 |
| DALL·E | 2.68 | 1.77 | 2.13 | 2.36 | 2.31 | 1.94 | 2.07 |
| Average | 2.95 | 2.39 | 2.35 | 2.56 | 2.54 | 2.35 | 2.21 |
| Sample Size | 48 | 64 | 56 | 80 | 40 | 64 | 52 |

Table 3: Correctness evaluation within each understanding category (compile errors are penalized with score 0).

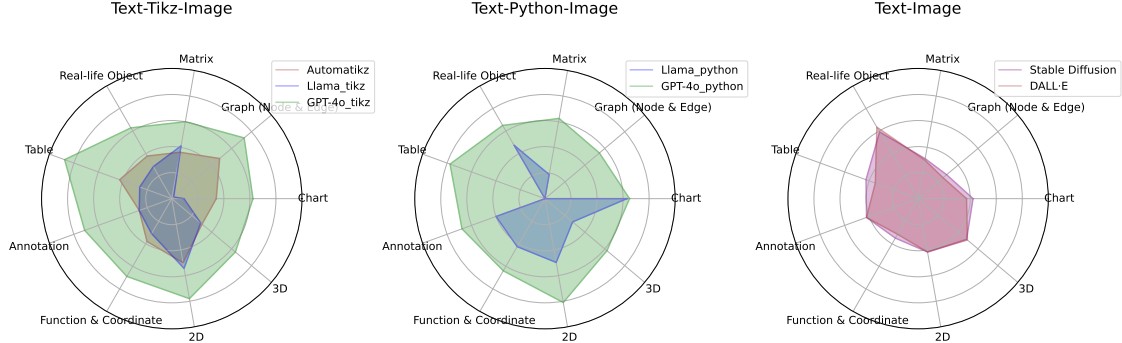

Figure 4: Generation performance of models on different object types. The same scale is used for three radar bars, with the center as correctness score 0, and the outermost circle as 5.

In contrast, for the image generation models STABLE DIFFUSION and DALL·E, numerical comprehension poses the greatest challenge (Figure 3). Both models score between below 1.8 for numerical understanding, substantially lower than their scores for attribute understanding (∼2.7) and spatial understanding (above 2.0). This indicates an interesting discrepancy between model types.

Due to their weakness in spatial understanding, tasks that involve combined understanding types—including numerical & spatial understanding, as well as numerical & spatial & attribute understanding—also tend to receive lower scores. Both GPT-4o_python and GPT-4o_tikz record their lowest scores when addressing prompts that require all three understanding types, in comparison to prompts focused on individual understanding types.

| Model | 2D shape | 3D shape | Chart | Graph theory | Matrix | Real-life object | Table | Annotation | Function& Coordinate |
|---|---|---|---|---|---|---|---|---|---|
| Automatikz | 2.50 | 1.52 | 1.71 | 2.40 | 1.81 | 1.89 | 2.13 | 1.33 | 1.90 |
| Llama_tikz | 2.72 | 1.45 | 0.47 | 0.10 | 2.06 | 1.42 | 1.13 | 1.33 | 1.55 |
| GPT-4o_tikz | 3.90 | 3.19 | 3.12 | 3.63 | 3.00 | 3.14 | 4.38 | 3.56 | 3.45 |
| Llama_python | 2.49 | 1.39 | 3.15 | 0.00 | 0.94 | 2.37 | 0.00 | 2.00 | 2.13 |
| GPT-4o_python | 4.05 | 3.11 | 3.25 | 2.73 | 3.13 | 3.25 | 3.88 | 3.39 | 3.20 |
| Stable Diffusion | 2.08 | 2.43 | 2.11 | 1.43 | 1.56 | 2.96 | 2.13 | 2.11 | 1.75 |
| DALL·E | 2.08 | 2.47 | 1.86 | 1.25 | 1.50 | 3.17 | 1.75 | 2.11 | 1.58 |
| Average | 2.83 | 2.22 | 2.24 | 1.65 | 2.00 | 2.60 | 2.20 | 2.26 | 2.22 |
| Sample Size | 162 | 97 | 54 | 20 | 8 | 38 | 4 | 9 | 20 |

Table 4: Object Complexity for Models: Correctness scores by object type.

**Object Categories** Table 4 presents the average correctness scores for each object category. The performance of different model types (Text-TikZ-Image, Text-Python-Image, and Text-Image) is visualized separately in Figure 4. In general, graph theory representation (e.g. nodes and edges in a binary tree or graph) poses great challenges for models, with an average score below 1.7 across all models, compared to above 2.0 the remaining object categories.

As illustrated in Figure 4, GPT-4o is the top-performing model across all object types, with both TikZ and Python-generated images consistently achieving the highest scores within each category. However, GPT-4o shows slightly lower performance in generating matrices (TikZ score: 3.00), 3D shapes (Python score: 3.11), and real-life object models (TikZ score: 3.14).

| Criteria | Correctness | | | | Relevance | | | | Scientific style | | | |
|---|---|---|---|---|---|---|---|---|---|---|---|---|
| Language | EN | DE | ZH | FA | EN | DE | ZH | FA | EN | DE | ZH | FA |
| Llama_tikz | 1.88 | 1.48 | 1.50 | 1.23 | 2.18 | 1.78 | 2.10 | 1.68 | 2.78 | 2.23 | 2.80 | 2.90 |
| GPT-4o_tikz | 3.85 | 4.03 | 3.98 | 3.68 | 4.03 | **4.23** | **4.60** | 3.98 | 4.10 | **4.43** | 4.40 | 3.98 |
| OpenAI-o1_tikz | **4.43** | 3.68 | 3.83 | **4.05** | **4.45** | 3.80 | 4.10 | **4.18** | 4.40 | 3.88 | 4.03 | **4.05** |
| Llama_python | 2.53 | 1.35 | 1.75 | 1.78 | 2.70 | 1.53 | 2.00 | 1.90 | 3.20 | 2.50 | 3.10 | 3.30 |
| GPT-4o_python | 3.38 | **4.15** | **4.13** | 3.48 | 3.35 | 4.18 | 4.23 | 3.35 | 3.88 | 4.50 | **4.83** | 3.85 |
| OpenAI-o1_python | 4.28 | 3.45 | 4.10 | 3.60 | 4.10 | 3.45 | 3.93 | 3.60 | **4.50** | 4.08 | 4.30 | **4.05** |
| Qwen2.5_python | 3.10 | 2.30 | 2.05 | 2.40 | 3.08 | 2.48 | 2.25 | 2.53 | 3.70 | 3.43 | 3.28 | 3.68 |
| DALL-E | 1.98 | 2.15 | 1.83 | 1.93 | 1.88 | 2.03 | 2.03 | 2.00 | 1.40 | 1.58 | 1.53 | 1.50 |
| Average | 3.18 | 2.82 | 2.89 | 2.77 | 3.22 | 2.93 | 3.15 | 2.90 | 3.49 | 3.33 | 3.53 | 3.41 |

Table 5: Multilingual performance of overall generations. The best is highlighted in bold per metrics per language.

LLAMA 3.1 8B (in both TikZ and Python code) clearly struggles with the representation of graph theory structures, such as nodes and edges, with a correctness score close to 0. Similarly, it performs poorly in table generation, scoring 1.13 with TikZ and 0 with Python. The largest discrepancy between TikZ and Python output is observed in chart generation, where TikZ achieves a score of 0.47, while Python scores 3.15. In contrast, TikZ outperforms Python in matrix generation, with correctness scores of 2.06 and 0.94, respectively. This may be attributed to the availability of libraries with Python and TikZ code. Matplotlib in Python is frequently used for chart and plot representation, while the usage of matrix presentation with proper math format is rare.

AUTOMATIKZ shows the lowest scores for annotation (1.33) and 3D geometric shape generation (1.52) across all object types. It performs best in 2D geometric shape generation (2.52), though it still lags behind GPT-4O, which scores above 3.

Figure 4 reveals that code-generated images are of higher quality for 2D geometric shapes compared to 3D shapes, while visual models exhibit the opposite trend. STABLE DIFFUSION and DALL·E perform best in real-life object modeling, with scores of 3.12 and 3.24, respectively, and in 3D geometric shape generation, with scores of 2.43 and 2.47.

| Code | Correctness | Relevance | Scientific style | Error rate |
|---|---|---|---|---|
| TikZ | 2.64 | 2.81 | 3.18 | 0.19 |
| Python | 2.81 | 2.97 | 3.56 | 0.17 |

Table 6: Comparison of TikZ code performance and Python code performance: scores averaged from model GPT-4O and LLAMA 3.1 8B. Compile errors are penalized with a score 0.

**Multilingual Evaluation** We further evaluate model performance across diverse languages. Due to the high annotation costs, we sample 20 instructions and translate them into Chinese, German and Farsi by native language co-authors. Each prompt is derived from a unique template to ensure diversity. Moreover, these instructions are curated to encompass all understanding dimensions: the single dimension of spatial, attribute, or numerical understanding; combinations of two dimensions (e.g., prompts requiring both attribute and numerical understanding); and prompts requiring all three understanding dimensions. We then feed the 20 prompts to all models except for AUTOMATIKZ, which is our only model fine-tuned on English TikZ data, and STABLE DIFFUSION. We additionally include the recently released OpenAI o1-preview, which emphasizes its science, coding, and mathematics capabilities (OpenAI, 2024), as well as Qwen2.5-Coder-7B-Instruct.[13]

Results are shown in Table 5. Interestingly, English does not always lead to the best results on average. While correctness and relevance is highest with English prompts with a margin of 0.29 and 0.07 ahead of Chinese, outputs generated from Chinese prompts are better in terms of scientificness. Farsi is worst on average. Model wise, LLAMA 3.1 8B becomes considerably worse regarding correctness and relevance in languages other than English, while GPT-4O often even perform better in non-English languages. Remarkably, OpenAI o1-preview is better than GPT-4O in English (and regarding its maximum scores), often by a considerable margin (e.g., 4.28 vs. 3.38 in correctness for English input with python output), but performs substantially worse in non-English languages (e.g., 3.45 vs. 4.15 in German with python output), except for Farsi. Qwen2.5-Coder-7B-Instruct demonstrates superior performance in English across all three evaluation dimensions, although Qwen generally exhibits robustness when responding to Chinese queries (Bai et al., 2023).

**Output code difference** As shown in Table 6, Python code generation outperforms TikZ output across all three evaluation criteria: Correctness, Relevance and Scientific style. Furthermore, the Python code output exhibits a lower

---

[13]We only present the results of Qwen2.5-Coder-7B-Instruct in text-python-image settings in Table 5 as its generated TikZ code frequently contains rendering errors that prevent successful conversion into images.

compile error rate (0.17 for Python vs. 0.19 for TikZ). The disparity may be attributed to the richer resources of available Python code in the training data for LLMs.

**Qualitative Analysis**   We diagnose some issues by closely examining the generated output images, specifically highlighting problems that point to areas for future improvement in the models. For instance, the generated images from some models reveal a lack of physics knowledge. In cases requiring an image of liquid in a container (Figure 5), the liquid is often placed incorrectly, not at the bottom of the container. This issue occasionally occurs with AUTOMATIKZ and LLAMA 3.1 8B, but it is not observed in models like GPT-4O, STABLE DIFFUSION and DALL·E.

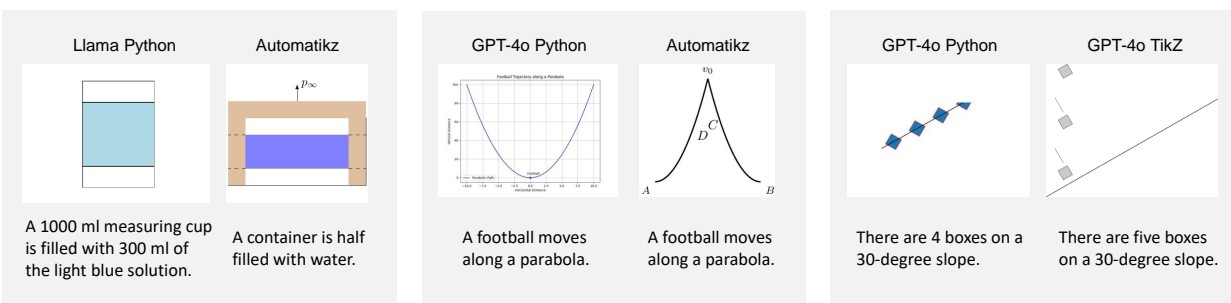

Figure 5: Incorrect model outputs can often be attributed to a lack of empirical, physical world knowledge.

A challenging scenario for most models is generating an object moving along a parabolic path. GPT-4O and LLAMA 3.1 8B occasionally depict a correct downward-opening parabola, but upward-opening parabolas also exist in their generation, indicating a lack of understanding of the trajectory of how an object moves. Another common issue across models is their difficulty in generating images that depict "boxes placed on a slope at a specific angle". Although GPT-4O sometimes manages to generate the correct image, their performance is inconsistent. This suggests a lack of understanding of the interaction between gravity and the support surface, as well as difficulty positioning objects at the correct angle on a 2D plane.

# 6   CONCLUDING REMARKS

Our study presents the first comprehensive evaluation of multimodal LLMs for scientific image generation, using our novel `ScImage` benchmark. Our assessment reveals both significant progress and persistent challenges in the field. While models like GPT-4O sometimes demonstrate proficiency in tasks involving individual dimensions of understanding (spatial, numeric, or attribute-based in isolation), all evaluated models struggle with complex tasks requiring combined understanding. On average, even GPT-4O performs below 4 on correctness on our benchmark. For example, due to its lack of world knowledge or an inability to correctly plan how a 3D object should be presented, GPT4 sometimes has difficulty arranging objects correctly in a two-dimensional image. Code based models have difficulties especially with spatial understanding, while image based models struggle the most with numeric understanding.

We find that code-based outputs generally outperform direct image generation in producing scientifically styled images. However, performance varies considerably across different object types and languages, highlighting the need for more robust and consistent modeling approaches in the scientific domain. These findings underscore the importance of continued research to enhance multimodal LLMs' capabilities in scientific image generation. By providing a standardized benchmark and detailed analysis, `ScImage` aims to drive progress in this critical area, supporting more efficient scientific communication and accelerating cross-disciplinary research.

As multimodal LLMs evolve, their potential to revolutionize scientific content generation remains an exciting frontier in AI research. Future work should focus on improving models' ability to handle complex, multi-dimensional reasoning tasks and ensure consistent performance across diverse scientific domains and languages. As science serves humanity and should be accessible by everyone to foster diversity and inclusion, this concerns particularly *open-source* models which can be considered at best mediocre for the tasks sketched in `ScImage`, with performances that substantially lag behind closed-source proprietary models like GPT-4.

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

## A    BENCHMARK SURVEY

| Dataset | Size | Sci-domain | Input | Output | Type of challenges |
|---|---|---|---|---|---|
| **STVQA** Biten et al. (2019) | ~23k | ✗ | image + question | answer to the question | text identification, recognation and reasoning |
| **TextVQA** Singh et al. (2019) | ~45k questions ~28k images | ✗ | image + question | answer to the question | text identification, recognation and reasoning |
| **EXAMS-V** Das et al. (2024) | ~20.9k | ✓ | image + questions | answer to the question | text identification, exam question reasoning |
| **MMVP** Tong et al. (2024) | 300 | ✗ | image + question | answer to the question | images with similar CLIP embeddings despite visual distinctions. |
| **MMMU** Yue et al. (2023) | 11.5K | ✓ | image + question | answer to the question | knowledge and reasoning of college exams |
| **science QA** Lu et al. (2022) | 21K | ✓ | image + question | answer + explanation (CoT) | scientific problem and image reasoning |
| **MathVista** Lu et al. (2024b) | ~6k | ✓ | image + question | answer to the question | math problem solving |
| **SciBench** Wang et al. (2024a) | 869 | ✓ | image + question | answer to the question | college level problem solving |
| **ArXivQA** Li et al. (2024a) | 100K | ✓ | image + question | answer to the question | GPT-4V generated questions for arXiv paper figures |
| **VGbench** Zou et al. (2024) | 4219 | ✓ | image + question | answer to the question | object category, color, object function, position, etc. |
| **CharXiv** Wang et al. (2024b) | 2323 | ✓ | Image + question | answer to the question | chart understanding and reasoning |
| **CONTEXTUAL** Wadhawan et al. (2024) | 506 | ✗ | image + question | answer to the question | image reasoning (avoid textual recognition or reasoning from language models) |
| **MMTBench** Ying et al. (2024) | 32k | ✗ | image + question | answer to the question | visual recognition, localization, OCR, counting, 3D perception, temporal understanding, et al. |
| **R-Bench** Wu et al. (2024) | 4500 images ~11.6k questions | ✗ | image + question | answer to the question | hallucination test: object reasoning, relationship (between objects) reasoning |
| **MM_Vet** Yu et al. (2024) | 187 imaegs 205 questions | ✗ | image + question | answer to the question | object recognition, spatial awareness, knowledge reasoning, math capability OCR, text generation |
| **Paper2Fig** Rodriguez et al. (2023) | 100k | ✓ | caption | image | scientific image generation |
| **Datikz** Belouadi et al. (2024a) | 120k | ✓ | caption | image | scientific image generation with TikZ code |
| **VGbench** Zou et al. (2024) | 5845 | ✓ | caption | Image | generate SVG, TikZ, and Graphviz images |
| **T2I-CompBench** Huang et al. (2023) | 6k | ✗ | caption | image | attribute binding, object relationships, complex composition |
| **PAINTSKILLS** Cho et al. (2023) | ~65k | ✗ | caption | image | object recognition, object counting spatial relation, etc |
| **HEIM** Lee et al. (2024) | - | ✗ | caption | image | reasoning, knowledge, multilinguality, etc |
| **SciFIBench** Roberts et al. (2024) | 1000 | ✓ | multiple choice question: select a correct caption for a figure or select a correct image for a given caption. | | ability to match image and caption |
| **MMSCI** Li et al. (2024b) | 742k figures | ✓ | scientific figure and caption pairs. (aim: alignment of figure-caption pairs) | | scientific figure understanding, figure-caption alignment |
| **ChartMimic** Shi et al. (2024) | 1000 | ✓ | chart image + instruction | Python code | chart image to Python code conversion |
| **Wino-ground** Thrush et al. (2022) | 1600 | ✗ | text-image pairs (aim: match the correct text-image pairs given two captions and two images) | | reasoning of objects and relationship difference by swapping words in 2 captions |

Table 7: Summary of challenging benchmarks in visual reasoning and text-to-image generation.

## B  GENERATION INSTRUCTION

| Generation Mode | Final Prompt |
|---|---|
| Text-image | "Please generate a scientific figure according to the following requirements: `{generation query}`". |
| Text-code-image | "Please generate a scientific figure according to the following requirements: `{generation query}`. Your output should be in [Python/Tikz] code. Do not include any text other than the [Python/Tikz] code." |

Table 8: Auxiliary instruction for constraining the model to generate scientific graphs in corresponding mode.

## C  GENERATION QUERY

| Object Type | Query Template | Attribution | Generation Query | Understanding Dimension |
|---|---|---|---|---|
| 2D shape | A/An {object} with a/an {color} border. | circle, blue | A circle with a blue border. | Attribute |
| 3D shape | Two {color} spaces divide a/an {3D-object} into four parts. | blue, pyramid | Two blue spaces divide a pyramid into four parts. | Attribute, Numeric |
| Chart | In a bar chart, a/an {color-1} bar is to the right of a/an {color-2} bar, and the leftmost bar is the {tallest/shortest} in the chart. | blue, orange, tallest | In a bar chart, a blue bar is to the right of an orange bar, and the leftmost bar is the tallest in the chart. | Attribute, Numeric, Spatial |
| Graph theory representation | A binary tree with a total of {number} nodes. | 12 | A binary tree with a total of 12 nodes. | Numeric |
| Matrix | A {number_1}-by-{number_2} matrix. | 6, 3 | A six-by-three matrix. | Numeric |
| Real-life object | There are {number-1} boxes on a {number-2} degree slope. | 4, 30 | There are 4 boxes on a 30 degree slope. | Numeric, Spatial |
| Table | A table with {number-1} rows and {number-2} columns. The row index is marked in the first column. | 3, 5 | A table with three rows and five columns. The row index is marked in the first column. | Attribute, Numeric, Spatial |
| Annotation | The English text (the name of the object) is {preposition} the {object}. | to the left of, triangle | The English text (the name of the object) is to the left of the triangle. | Spatial |
| Function & Coordinate | y = {function} and its inverse function. | 3^x | y = 3^x and its inverse function. | Numeric |

Table 9: Illustration of constructing generation queries.

# D   CODE OUTPUT

**GPT-4o Tikz Code**

```
\documentclass{standalone}
\usepackage{tikz}
\begin{document}

\begin{tikzpicture}

% Green cone
\fill[green] (0,0) --
    (-1,-2) -- (1,-2) --
    cycle;

% Brown ellipsoid
\shade[ball color=brown]
    (3,-1) ellipse (1.5 and
    1);

\end{tikzpicture}

\end{document}
```

**Llama Tikz Code**

```
\documentclass{standalone}
\usepackage{tikz}
\begin{document}

\begin{tikzpicture}[scale=2]
\draw[fill=green!80] (0,0)
    -- (1,0) arc (0:180:1) --
    cycle;
\draw[fill=brown!80] (1,0)
    -- (1.5,0.5) -- (1,1) --
    cycle;

\begin{scope}[shift={(1,0)}]
\draw[fill=brown!80] (0,0)
    -- (0.5,0) -- (0.5,0.5)
    -- cycle;
\end{scope}

\end{tikzpicture}

\end{document}
```

Figure 6: TikZ code snippets of GPT-4o and Llama models for generating a cone and ellipsoid.

**GPT-4o Python Code**

```python
import matplotlib.pyplot as plt
import numpy as np
from mpl_toolkits.mplot3d.art3d import
    Poly3DCollection
# Create figure and 3D axis
fig = plt.figure()
ax = fig.add_subplot(111,
    projection='3d')
# Define a function to create a cone
def create_cone(ax, radius, height,
    color):
    u = np.linspace(0, 2 * np.pi, 100)
    x = radius * np.cos(u)
    y = radius * np.sin(u)
    z = np.zeros_like(x)
    verts = [list(zip(x, y, z))]
    verts.append([(0, 0, height)])  #
        Apex of the cone
    faces = Poly3DCollection(verts,
        color=color, alpha=0.6)
    ax.add_collection3d(faces)
# Create a function to plot an ellipsoid
def create_ellipsoid(ax, rx, ry, rz,
    color):
    u = np.linspace(0, 2 * np.pi, 100)
    v = np.linspace(0, np.pi, 100)
    x = rx * np.outer(np.cos(u),
        np.sin(v))
    y = ry * np.outer(np.sin(u),
        np.sin(v))
    z = rz * np.outer(np.ones_like(u),
        np.cos(v))
    ax.plot_surface(x, y, z,
        color=color, alpha=0.6)
# Parameters for the shapes
cone_radius = 2
cone_height = 5
ellipsoid_rx = 3
ellipsoid_ry = 1.5
ellipsoid_rz = 2
# Create and plot the cone
create_cone(ax, cone_radius,
    cone_height, 'green')
# Create and plot the ellipsoid
create_ellipsoid(ax, ellipsoid_rx,
    ellipsoid_ry, ellipsoid_rz, 'brown')
# Set plot limits
ax.set_xlim([-5, 5])
ax.set_ylim([-5, 5])
ax.set_zlim([0, 10])
# Display the plot
plt.show()
```

**Llama Python Code**

```python
import matplotlib.pyplot as plt
from mpl_toolkits.mplot3d import Axes3D
import numpy as np

# Create a figure and a 3D axis
fig = plt.figure(figsize=(8, 6))
ax = fig.add_subplot(111,
    projection='3d')
# Define the parameters of the cone
cone_height = 5
cone_radius = 1
# Define the parameters of the ellipsoid
ellipsoid_x_radius = 2
ellipsoid_y_radius = 1.5
ellipsoid_z_radius = 1
# Generate the vertices of the cone
u = np.linspace(0, 2 * np.pi, 100)
v = np.linspace(0, cone_height, 100)
x = (cone_radius * np.outer(np.cos(u),
    np.sin(v)))
y = (cone_radius * np.outer(np.sin(u),
    np.sin(v)))
z = (cone_height * np.ones(np.size(v)))
    - (cone_radius *
    np.outer(np.ones(np.size(u)),
    np.cos(v)))
# Plot the cone
ax.plot_surface(x, y, z, color='green',
    alpha=0.5)
# Generate the vertices of the ellipsoid
u = np.linspace(0, 2 * np.pi, 100)
v = np.linspace(0, 2 * np.pi, 100)
x_ellipsoid = (ellipsoid_x_radius *
    np.outer(np.cos(u), np.sin(v)))
y_ellipsoid = (ellipsoid_y_radius *
    np.outer(np.sin(u), np.sin(v)))
z_ellipsoid = (ellipsoid_z_radius *
    np.outer(np.ones(np.size(u)),
    np.cos(v)))
# Plot the ellipsoid
ax.plot_surface(x_ellipsoid,
    y_ellipsoid, z_ellipsoid,
    color='brown', alpha=0.5)
# Set the limits of the axes
ax.set_xlim(-3, 3)
ax.set_ylim(-3, 3)
ax.set_zlim(0, 6)

# Show the plot
plt.show()
```

Figure 7: Python code snippets of GPT-4o and Llama models for generating a cone and ellipsoid.

# E    IMAGE EXAMPLES

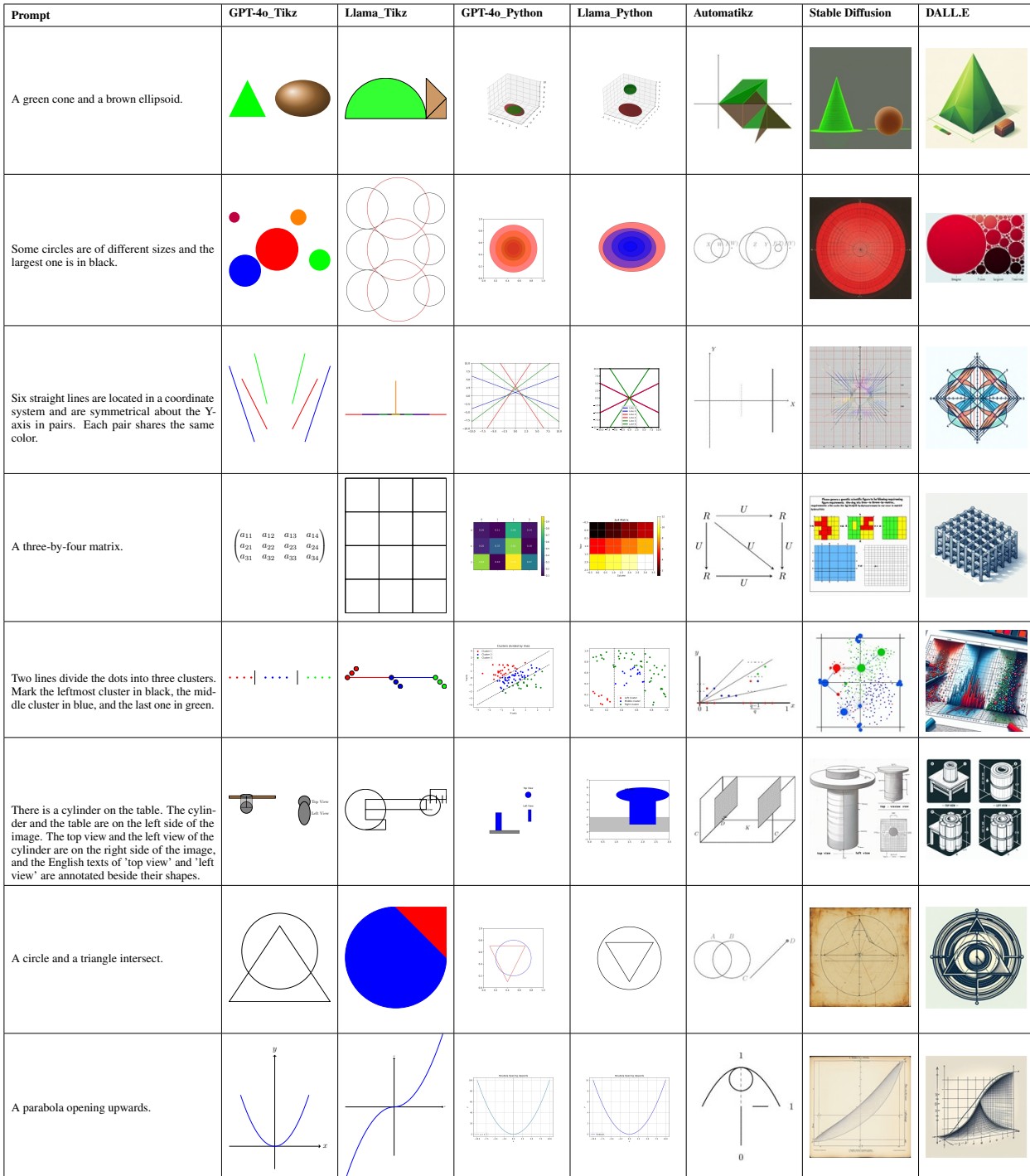

Table 10: **Comparison of Generated Images by Different Models from Various Prompts:** Each column in this table presents the side-by-side comparison of images generated by the different models in response to the corresponding prompts. Results for each prompt are shown in each row, demonstrating the diversity of model approaches and styles. Image outputs for the first four columns are generated through the models' code, while the others are generated directly by prompting the models.

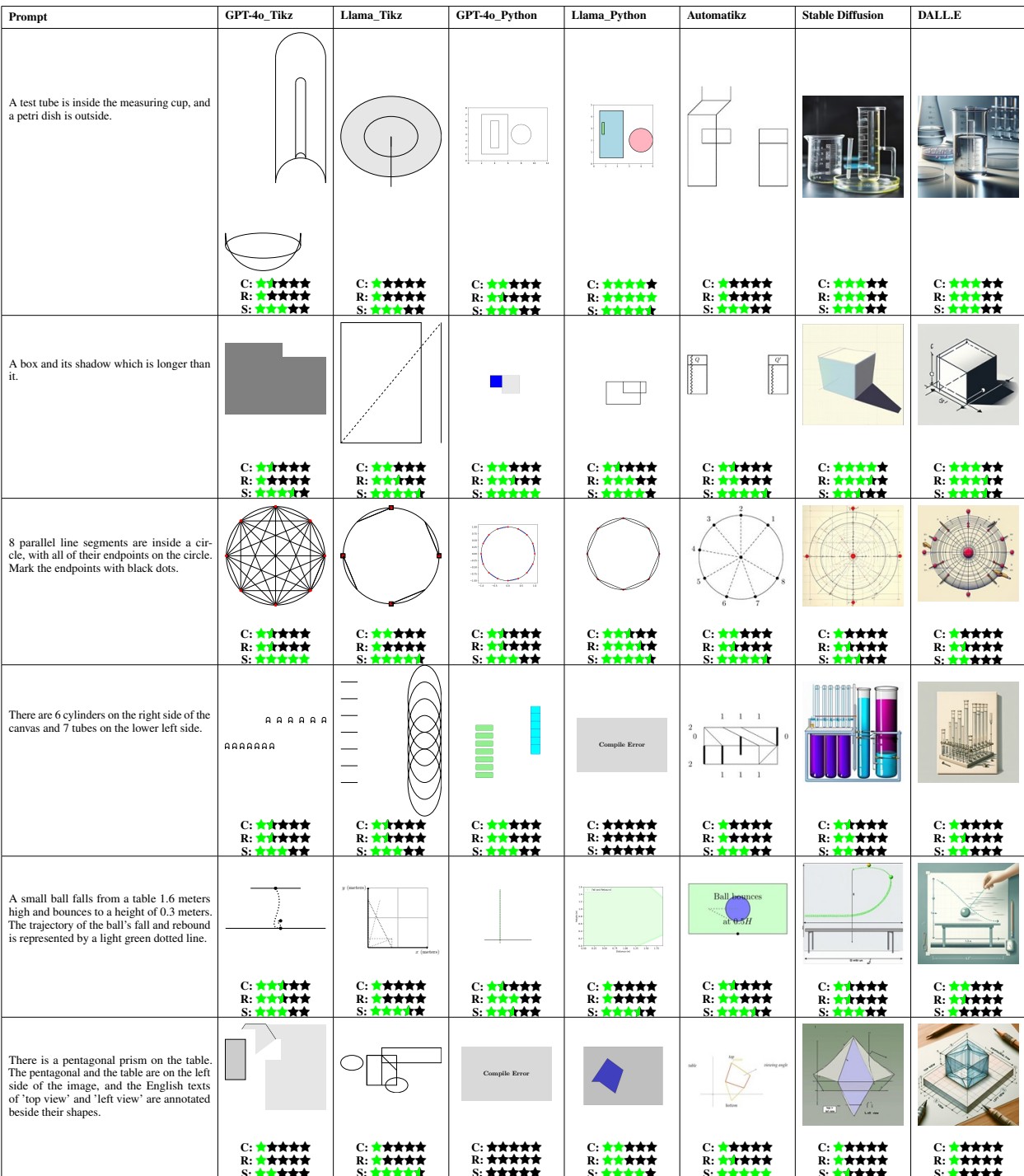

Table 11: **Failure Cases of Different Models:** The examples give selected cases where both GPT4o models have an average correctness score ≤ 2.5. The last four rows depict instances where all models have correctness scores ≤ 2.5. Scores across Correctness (C), Relevance (R), and Scientific Style (S) are illustrated as star ratings.

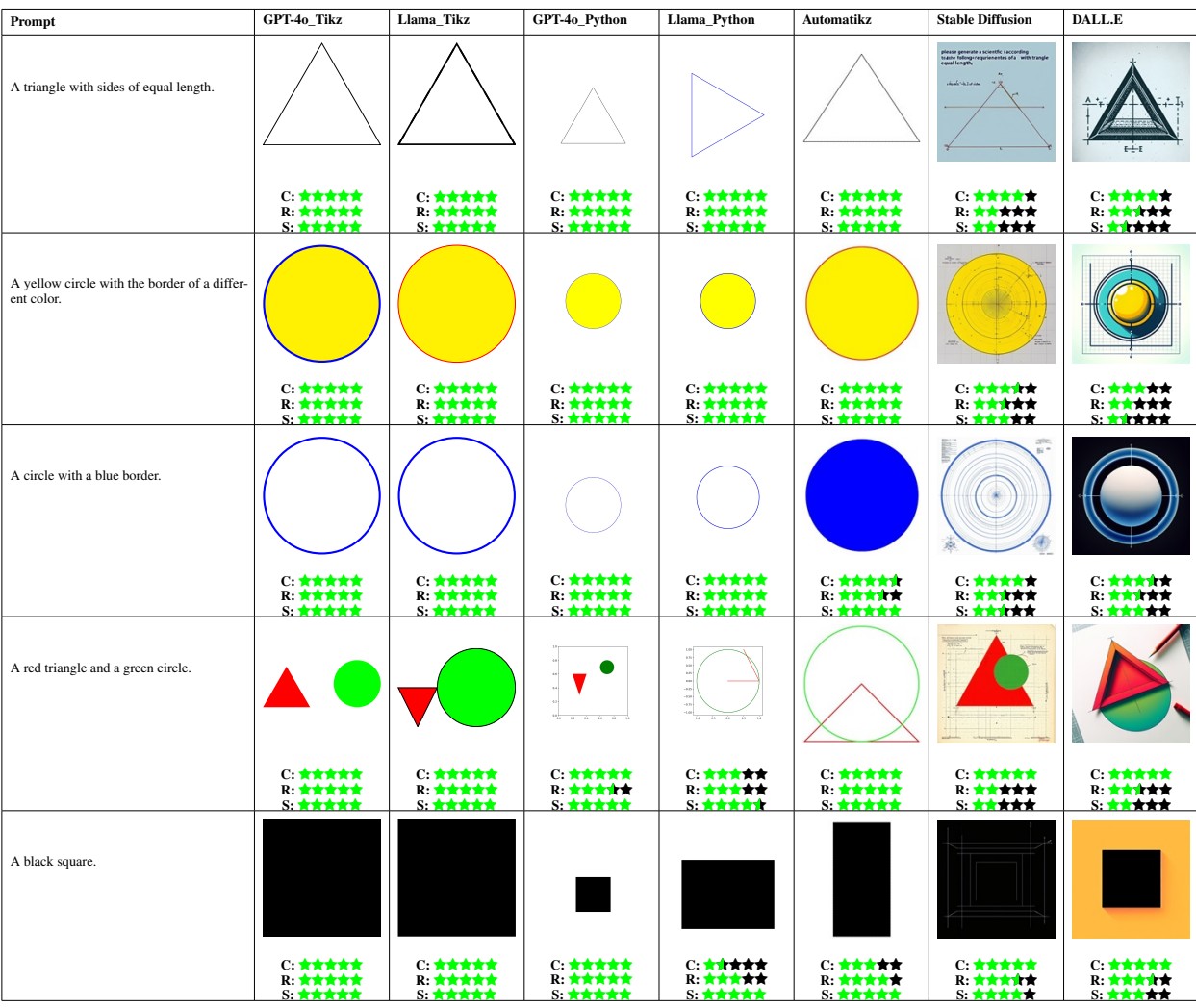

Table 12: **Gold Standard Examples:** The examples give selected instances where models have an average correctness score $\geq 4$ in the majority of cases. Scores across Correctness (C), Relevance (R), and Scientific Style (S) are illustrated as star ratings.

## F    EVALUATION GUIDELINE

### F.1    CORRECTNESS

| Score | Description |
|-------|-------------|
| 5 | The image fully meets all the requirements with no mistakes. |
| 4 | The image meets the key requirements, with only minor mistakes. |
| 3 | The image meets some or half of the requirements, with some mistakes. |
| 2 | The image meets only a few of the text's requirements and contains serious mistakes. |
| 1 | The image fails to meet the requirements of the text. |
| 0 | No image content or compile error. |

Table 13: Correctness Scoring Guideline

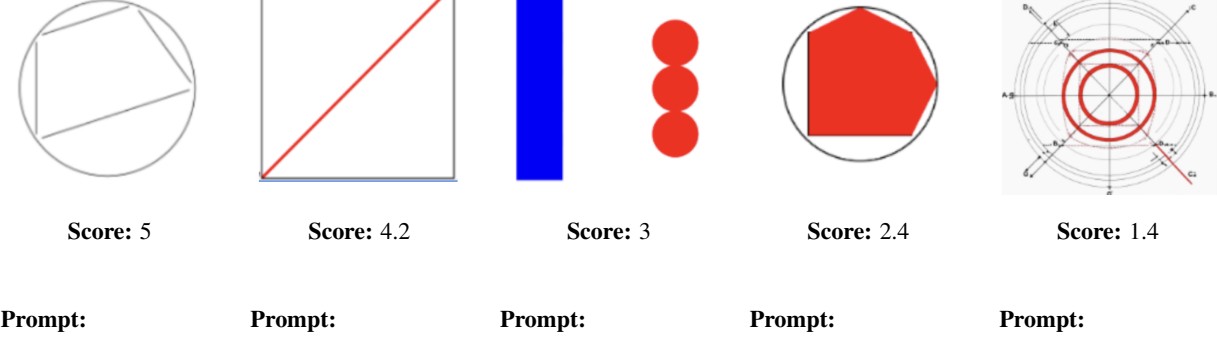

| **Score:** 5 | **Score:** 4.2 | **Score:** 3 | **Score:** 2.4 | **Score:** 1.4 |
|---|---|---|---|---|
| **Prompt:** | **Prompt:** | **Prompt:** | **Prompt:** | **Prompt:** |
| There are 4 line segments inside a circle. | The diagonal of the square is red and very thick. | There are 5 squares on the left side of the canvas and 3 circles on the right side. | The intersection of the circle and the square is filled with red. | The intersection of the circle and the square is filled with red. |
| **Explanation:** | **Explanation:** | **Explanation:** | **Explanation:** | **Explanation:** |
| There are 4 line segments and they are inside a circle. | The diagonal of the output image is not visibly very thick. Other than that it's correct. | partially correct: The left graph shows a rectangle instead of 5 squares. The right-side graph is correct. | The red-marked intersection does not involve a square. However, there is a red intersection and a circle. | There is no square and no intersection of two graphs. (There is a circle and some red, but it's overall incorrectly displayed). |

Table 14: **Correctness Guideline Examples**
(Scores are averaged across multiple annotators)

## F.2 RELEVANCE

| Score | Description |
|-------|-------------|
| 5 | The image contains no redundant objects or features. |
| 4 | The image contains a few redundant objects or features but remains highly relevant to the text's requirements. |
| 3 | The image contains some redundant objects and some required elements. |
| 2 | The image contains more redundant objects than required elements. |
| 1 | The overall image is not relevant to the requirements. |
| 0 | No image content or compile error. |

Table 15: Relevance Scoring Guideline

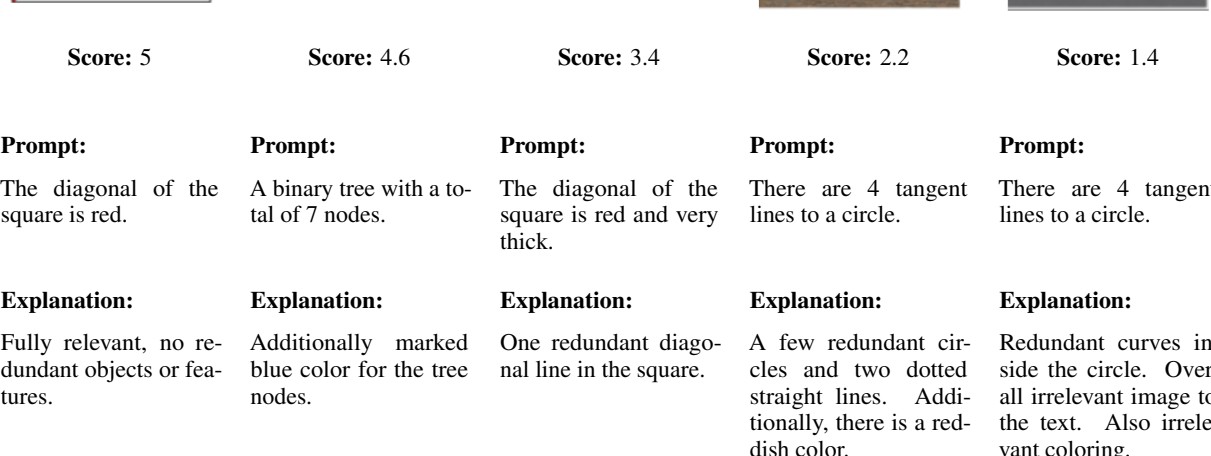

| **Score:** 5 | **Score:** 4.6 | **Score:** 3.4 | **Score:** 2.2 | **Score:** 1.4 |
|---|---|---|---|---|
| **Prompt:** | **Prompt:** | **Prompt:** | **Prompt:** | **Prompt:** |
| The diagonal of the square is red. | A binary tree with a total of 7 nodes. | The diagonal of the square is red and very thick. | There are 4 tangent lines to a circle. | There are 4 tangent lines to a circle. |
| **Explanation:** | **Explanation:** | **Explanation:** | **Explanation:** | **Explanation:** |
| Fully relevant, no redundant objects or features. | Additionally marked blue color for the tree nodes. | One redundant diagonal line in the square. | A few redundant circles and two dotted straight lines. Additionally, there is a reddish color. | Redundant curves inside the circle. Overall irrelevant image to the text. Also irrelevant coloring. |

Table 16: **Relevence Guideline Examples**
(Scores are averaged across multiple annotators)

### F.3 SCIENTIFICNESS

| Score | Description |
|-------|-------------|
| 5 | The image can be presented in a textbook or academic paper without any change. |
| 4 | The image has minor issues and requires minimal adjustments to be presented in a textbook or academic paper. |
| 3 | The image has serious issues in scientific style, including mismatched sizes, unsuitable positions, overlapping graphs or text, incomplete graphs, etc. |
| 2 | The image's style is not common in scientific settings. |
| 1 | The overall image is more suitable in a lifestyle context and is not appropriate for scientific demonstration. |
| 0 | No image content or compile error. |

Table 17: Scientificness Quality Scoring Guideline

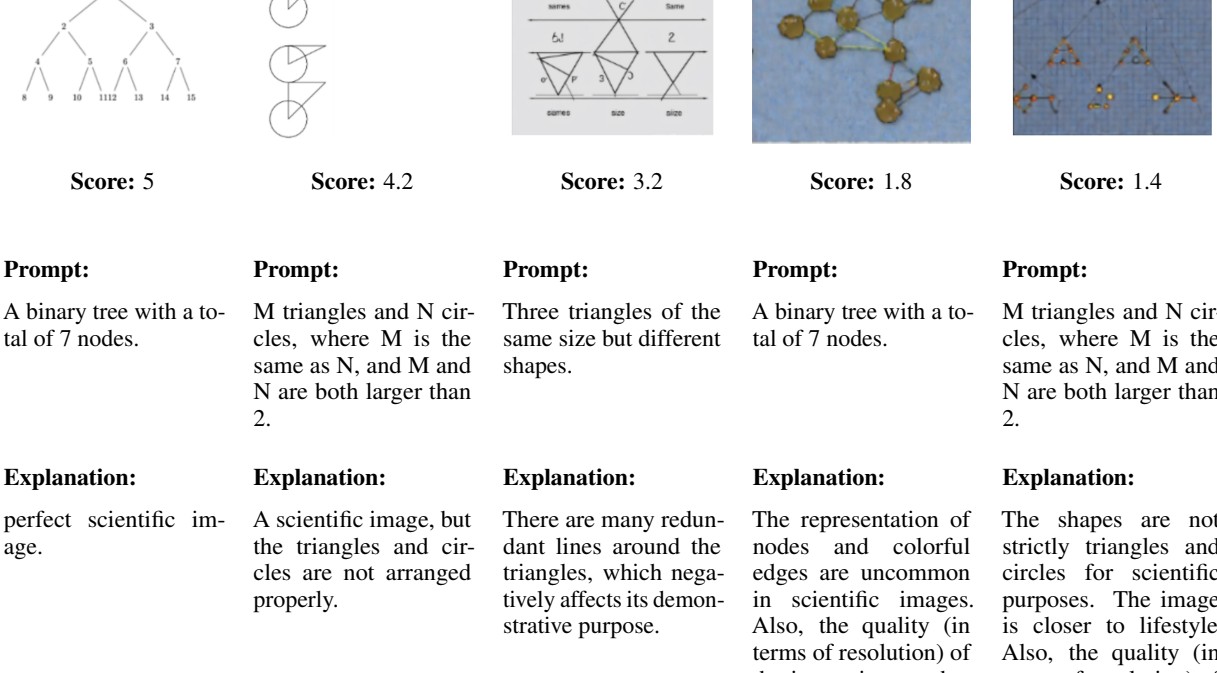

| Score: 5 | Score: 4.2 | Score: 3.2 | Score: 1.8 | Score: 1.4 |
|-----------|------------|------------|------------|------------|
| **Prompt:** | **Prompt:** | **Prompt:** | **Prompt:** | **Prompt:** |
| A binary tree with a total of 7 nodes. | M triangles and N circles, where M is the same as N, and M and N are both larger than 2. | Three triangles of the same size but different shapes. | A binary tree with a total of 7 nodes. | M triangles and N circles, where M is the same as N, and M and N are both larger than 2. |
| **Explanation:** | **Explanation:** | **Explanation:** | **Explanation:** | **Explanation:** |
| perfect scientific image. | A scientific image, but the triangles and circles are not arranged properly. | There are many redundant lines around the triangles, which negatively affects its demonstrative purpose. | The representation of nodes and colorful edges are uncommon in scientific images. Also, the quality (in terms of resolution) of the image is not what one would expect in modern science. | The shapes are not strictly triangles and circles for scientific purposes. The image is closer to lifestyle. Also, the quality (in terms of resolution) of the image is not what one would expect in modern science. |

Table 18: **Scientificness Guideline Examples**
(Scores are averaged across multiple annotators)

## G  EVALUATION OF SUCCESSFULLY COMPILED IMAGES

| Models | Correctness | Relevance | Scientific Style | Compile Error Rate |
|---|---|---|---|---|
| Automatikz | 2.14 | 2.41 | 3.50 | **0.04** |
| Llama_tikz | 2.49 | 2.72 | 3.66 | 0.29 |
| GPT-4o_tikz | **3.84** | **4.02** | 4.10 | 0.09 |
| Llama_python | 2.92 | 3.52 | **4.42** | 0.28 |
| GPT-4o_python | 3.76 | 3.64 | 4.21 | 0.07 |
| Stable Diffusion | 2.19 | 2.09 | 1.96 | - |
| DALL·E | 2.16 | 2.00 | 1.55 | - |

Table 19: Overall Model Performance, averaged across all generated images (not including compile error cases), and two annotators.

| Criteria | Correctness | | | | Relevance | | | | Scientific style | | | |
|---|---|---|---|---|---|---|---|---|---|---|---|---|
| Language | EN | DE | ZH | FA | EN | DE | ZH | FA | EN | DE | ZH | FA |
| Llama_tikz | 2.34 | 2.68 | 2.14 | 1.75 | 2.72 | 3.23 | 3.00 | 2.39 | 3.47 | 4.05 | 4.00 | 4.14 |
| GPT-4o_tikz | 4.05 | 4.24 | 4.18 | 4.32 | 4.24 | 4.45 | 4.84 | 4.68 | 4.32 | 4.66 | 4.63 | 4.68 |
| GPT-o1_tikz | 4.66 | 4.59 | 4.50 | 4.76 | 4.68 | 4.75 | 4.82 | 4.91 | 4.63 | 4.84 | 4.74 | 4.76 |
| Llama_python | 3.88 | 2.08 | 2.19 | 1.97 | 4.15 | 2.35 | 2.50 | 2.11 | 4.92 | 3.85 | 3.88 | 3.67 |
| GPT-4o_python | 3.75 | 4.15 | 4.13 | 4.09 | 3.72 | 4.18 | 4.23 | 3.94 | 4.31 | 4.50 | 4.83 | 4.53 |
| GPT-o1_python | 4.28 | 3.83 | 4.32 | 4.00 | 4.10 | 3.83 | 4.13 | 4.00 | 4.50 | 4.53 | 4.53 | 4.50 |
| Qwen2.5_python | 3.10 | 2.30 | 2.05 | 2.40 | 3.08 | 2.48 | 2.25 | 2.53 | 3.70 | 3.43 | 3.28 | 3.68 |
| DALL-E | 1.98 | 2.15 | 1.83 | 1.93 | 1.88 | 2.03 | 2.03 | 2.00 | 1.40 | 1.58 | 1.53 | 1.50 |
| Average | 3.50 | 3.25 | 3.17 | 3.15 | 3.57 | 3.41 | 3.47 | 3.32 | 3.91 | 3.93 | 3.92 | 3.93 |

Table 20: Multilingual performance of compiled images (compile errors from generation are not included) The highest values across languages are highlighted in bold.

| Model | Attribute | Numerical | Spatial | Attribute & Numerical | Attribute & Spatial | Numerical & Spatial | Attribute & Numerical & Spatial |
|---|---|---|---|---|---|---|---|
| Automatikz | 2.70 | 2.01 | 1.80 | 2.35 | 2.04 | 2.17 | 1.88 |
| Llama_tikz | 2.83 | 2.75 | 2.15 | 2.56 | 2.45 | 2.39 | 2.33 |
| GPT-4o_tikz | 4.11 | 3.99 | 3.54 | 3.9 | 4.03 | 3.77 | 3.54 |
| Llama_python | 2.99 | 3.72 | 2.5 | 3.31 | 2.94 | 2.36 | 2.56 |
| GPT-4o_python | 4.12 | 4.05 | 3.81 | 3.74 | 3.93 | 3.44 | 3.33 |
| Stable Diffusion | 2.75 | 1.73 | 2.06 | 2.41 | 2.46 | 1.96 | 2.11 |
| DALL-E | 2.68 | 1.77 | 2.13 | 2.36 | 2.31 | 1.94 | 2.07 |

Table 21: Correctness evaluation within each understanding category (compile errors are not included).

| Model | 2D shape | 3D shape | Chart | Graph theory | Matrix | Real-life object | Table | Annotation | Function& Coordinate |
|---|---|---|---|---|---|---|---|---|---|
| Automatikz | 2.60 | 1.60 | 1.85 | 2.53 | 2.07 | 1.88 | 2.00 | 1.50 | 2.00 |
| Llama_tikz | 3.10 | 1.93 | 1.96 | 1.00 | 2.75 | 1.71 | 1.75 | 2.00 | 1.94 |
| GPT-4o_tikz | 4.13 | 3.40 | **4.11** | **4.26** | **4.00** | 3.28 | **3.56** | **3.88** | **3.63** |
| Llama_python | 3.01 | 2.21 | 3.95 | 0.00 | 2.50 | 2.90 | 1.50 | 3.33 | 2.50 |
| GPT-4o_python | **4.28** | **3.47** | 3.31 | 3.63 | 3.57 | **3.42** | 3.25 | **3.88** | 3.20 |
| Stable Diffusion | 2.08 | 2.43 | 2.11 | 1.43 | 1.56 | 3.12 | 1.94 | 1.63 | 1.75 |
| DALL·E | 2.08 | 2.47 | 1.86 | 1.25 | 1.50 | 3.24 | 2.13 | 2.13 | 1.58 |

Table 22: Correctness Evaluation within each object type (compile errors are not included). The column-wise highest is highlighted in bold.

# H    AUTOMATIC EVALUATION METHODS

We also test how well recent automatic text-to-image evaluation metrics correlate with our human judgements. In specific, we test the metrics CLIPScore (vit-base-patch16 and vit-large-patch14) (Hessel et al., 2021), ALIGNScore (Saxon et al., 2024) and PickScore (Kirstain et al., 2023), as well as the multimodel LLM based metric DSG (Cho et al., 2024) with Gemma-2-9B-SimPo (Team et al., 2024; Meng et al., 2024) for question generation. Table 23 shows the resulting Kendall correlations on a per-image granularity. The highest correlation is reached by PickScore that was trained on a large-scale dataset of human preference labels for generated images. However, the human Kendall correlations for this task, with $0.75$ (correctness), $0.68$ (relevance) and $0.62$ (scientificness) are much higher. This suggests the need for more suitable evaluation metrics in the domain of scientific text-to-image generation.

| | Correctness | Relevance | Scientificness |
|---|---|---|---|
| DSG (Gemma2) | 0.18 | 0.15 | 0.02 |
| CLIPScore | -0.00 | 0.01 | 0.04 |
| CLIPScore$_{Large}$ | 0.03 | 0.03 | 0.04 |
| AlignScore | 0.23 | 0.21 | 0.09 |
| PickScore | **0.26** | **0.23** | **0.15** |

Table 23: Kendall correlation between automatic metrics and human scores.

# I    LIMITATIONS

We note that AUTOMATIKZ may have been bad especially because it was trained on textual descriptions taken from captions of scientific papers, which may look substantially different from the instructions used in ScImage. For consistency, we did not develop model specific prompts, however. This holds more generally: while prompting is known to have a (sometimes substantial) effect on model performances (Mizrahi et al., 2024; Leiter et al., 2023), our study used one and the same prompt across all models.

One interesting avenue to explore in future work is the combination of heterogeneous LLMs, as we saw that models have complementary strengths and limitations. We finally note that sample sizes for some of the scientific objects considered and for the multilingual evaluation were comparatively small. This means the corresponding results need to be interpreted with caution.

Ethically, there is that risk that naive scientific users may place unwarranted trust in the output generated by some of the models, e.g., GPT-4O, without assessing whether the generated output confirms with their expectations or their prompted input. The human user has to take full responsibility for the outputs created by the models explored in our work.

