# OpenReview forum: "ScImage: How good are multimodal large language models at scientific text-to-image generation?"
_ICLR.cc/2025/Conference — ICLR 2025 Poster_

### Official Review · Reviewer_prga · 2024-11-02

**Soundness:** 2
**Presentation:** 3
**Contribution:** 3
**Rating:** 8
**Confidence:** 4

**Summary:**

This paper introduces a new benchmark, ScImage, which evaluate models abilities of scientific text-to-image generation, in both code-based and multimodal modes. This benchmark is multilingual, containing 4 languages. This benchmark emphasizes models' spatial understanding, numeric understanding, and attribute binding abilities. The authors use human evaluation for images generated by 5 models, where gpt-4o outperformed all other models.

**Strengths:**

- The authors introduce ScImage, which contributes to evaluation of models' scientific image generation capabilities.
- ScImage includes a wide range of tasks, which could provide comprehensive evaluation of models' image generation abilities.
- The authors provide detailed query construction process and templates, which could help creation of new queries.
- The authors provide example queries and example outputs of models, which facilitates understanding the benchmark.

**Weaknesses:**

- The authors use human evaluation to assess models' performances on this benchmark. It is good to know how humans rate the outputs, but it is less scalable if using human evaluation. If other people would like to test their models on this benchmark, they would need to also hire professional human evaluators, which would be expensive given that this benchmark contain more than 400 instances. It would be good if the authors could explore automated metrics that have high correlation with human evaluation results, or provide a crowdsource platform where human eval could be less costly, or use llm-as-a-judge methods to automatically evaluate model outputs.
- The authors provide limited number of models' results on this benchmark. It would be good to see how other open source models perform on this benchmark, such as

**Questions:**

- How should people evaluate other models' performances on this benchmark without needing human evaluators? For example, using image-image similarity between the generated model output with a oracle image for each task could provide a basic idea on how well the model output is. Additionally, llm-as-a-judge could potentially evaluate the model outputs as well.

---

> ### Author Response · Authors · 2024-11-22
>
> >The authors use human evaluation to assess models' performances on this benchmark. It is good to know how humans rate the outputs, but it is less scalable if using human evaluation. If other people would like to test their models on this benchmark, they would need to also hire professional human evaluators, which would be expensive given that this benchmark contain more than 400 instances. It would be good if the authors could explore automated metrics that have high correlation with human evaluation results, or provide a crowdsource platform where human eval could be less costly, or use llm-as-a-judge methods to automatically evaluate model outputs.
>
> Thanks a lot. We agree that having no good metric is an issue and we hope that we can produce one in the future (e.g., by fine-tuning on our human annotations). In the meantime, we have explored several recent SOTA metrics, and they all perform badly. And yes, we agree that the human evaluation is quite expensive, around 3000 USD in our case. Crowd-sourcing those evaluations may be an alternative, but previous research has shown that crowdworkers do not perform well for scientific tasks (Belouadi et al., 2024).  LLM-as-a-judge is the way modern metrics are built, but our findings implicate that they do not work well either.
>
> >The authors provide limited number of models' results on this benchmark. It would be good to see how other open source models perform on this benchmark, such as
>
> We thank the reviewer for this point, but want to point out that we evaluate multiple models on the benchmark. This includes textual models: Automatikz, GPT4o, GPTo1-preview, Llama 3.1 and visual models: Stable Diffusion and DALLE. We believe that GPT4o already constitutes the current upper bound and other models would not perform better. We also note that including more models increases the cost of human annotation. But we’re happy to test the reviewer’s favorite model if they outline it to us.
>
> >How should people evaluate other models' performances on this benchmark without needing human evaluators? For example, using image-image similarity between the generated model output with a oracle image for each task could provide a basic idea on how well the model output is. Additionally, llm-as-a-judge could potentially evaluate the model outputs as well.
>
> We agree that this is a problem but want to point out that this also emphasizes the value of our human annotation campaign. Image-to-image similarity would require a human reference image, which is very costly to produce for humans as well. Furthermore, image-to-image similarity is not well explored in science, and researchers can question the performance of evaluating graphs when science requires a high degree of precision, e.g., the precision of values ​​on a bar graph.  We do believe that our work exposes the need for better text-image evaluation metrics - via better multimodal LLMs - for the scientific domain in the future.
>
> We further point out that future research can train metrics on our human annotated data to obtain better metrics and by leveraging the images where models perform well, we can obtain an image reference dataset. We add this as a contribution and deliverable of our work.

---

> > ### Comment · Reviewer_prga · 2024-11-24
> >
> > Thank you for your responses! For me, the main concern is how other researchers evaluate their models on your benchmark.
> > Additionally, in the revised version, the caption of Table 7 overlapped with the main text.
> >
> > > In the meantime, we have explored several recent SOTA metrics, and they all perform badly.
> >
> > Could you elaborate on which metrics you explored, and their performances?
> >
> > > And yes, we agree that the human evaluation is quite expensive, around 3000 USD in our case.
> >
> > It would be nice to add this to the paper.
> >
> > > LLM-as-a-judge is the way modern metrics are built, but our findings implicate that they do not work well either.
> >
> > Could you clarify on what findings of yours implicate that LLM-as-a-judge doesn't work well?

---

> ### Author Response · Authors · 2024-11-24
>
> Thanks a lot for the quick reply!
>
> > Could you elaborate on which metrics you explored, and their performances?
>
> They are listed in Appendix G, especially Table 22, in the appendix. A summarizing description is given in l. 318ff.
>
> > It would be nice to add this to the paper.
>
> Thank you, it's already there: l. 313+314.
>
> > Could you clarify on what findings of yours implicate that LLM-as-a-judge doesn't work well?
>
> Yes, sure. The metric DSG (https://openreview.net/forum?id=ITq4ZRUT4a) for example is a modern LLM-based metric (a metric based on LLMs as judges). According to our Table 22 in the appendix it performs badly, along with all other metrics.
>
> > Thank you for your responses! For me, the main concern is how other researchers evaluate their models on your benchmark.
>
> That's a fair point. Let us point out several things:
>
> a) First, for now, we believe there are no strong automatic (text-to-image) metrics for our benchmark at all. So, for now, human involvement is indispensable, emphasizing the value of our human evaluation campaign. But we imagine that there will be soon some good automatic metrics (it's on our agenda for our next paper, for example).
>
> b) Secondly, for a subset of the benchmark we do have reference images: we can merely take all those images which are scored with 5 or with 4 across all input dimensions for at least one of our LLMs. By doing so, we have gold standard images for **2/3 of all of our prompts**.
>
> c) If the concern is so important for the reviewer, we could promise to add more references to a further subset (not sure if this can really be all prompts, but we'd give our best), e.g., by (i) searching images corresponding to our prompts on the web, (ii) doing extensive dialogical interaction with GPT4o, or (iii) simply drawing the image ourselves. While we imagine it will cost another 1000+ USD to do so, it is feasible. We could certainly have this by the time the CR is due.
>
> > Additionally, in the revised version, the caption of Table 7 overlapped with the main text.
>
> Oops, thanks for pointing this out. We will certainly fix this in the next update.

---

> > ### Comment · Reviewer_prga · 2024-11-24
> >
> > Thank you for your prompt response. Your comments have largely addressed my concerns. I have adjusted my score, trusting that the promised revisions will be reflected in the final version.

---

> ### Author Response · Authors · 2024-11-25
>
> Thanks a lot for your quick reply and the positive evaluation!
> Your review was much appreciated.

---

### Official Review · Reviewer_4MC8 · 2024-11-03

**Soundness:** 2
**Presentation:** 2
**Contribution:** 2
**Rating:** 3
**Confidence:** 4

**Summary:**

This work presents a novel benchmark dataset designed to evaluate the multimodal capabilities of large language models (LLMs) in generating scientific images from textual descriptions. The dataset encompasses a wide variety of formats, including 2D/3D shapes, charts, graph theory representations, matrices, tables, and more, with a focus on assessing spatial, numerical, and attribute comprehension. The evaluation process involved 11 scientists who reviewed the performance of models such as GPT-4, LLaMA, and Stable Diffusion.

However, the dataset primarily consists of query templates, which limits its applicability to other models, as the evaluation heavily relies on human (scientist) assessments. Furthermore, the technical details provided are not sufficiently rigorous, with multimodal models and Stable Diffusion collectively categorized as large language models, resulting in imprecision and lack of clarity. Additionally, several relevant prior works are inadequately cited or discussed, which detracts from the comprehensiveness of the study.

**Strengths:**

1. This dataset includes a wider variety of image types compared to previous work that primarily focused on charts and bar graphs.
2. The evaluation was conducted by 11 scientists, ensuring high-quality assessment. However, this dependence on human evaluation limits the benchmark dataset's scalability for testing additional models.
3. The study also evaluates various textual description languages and coding languages/packages, making the analysis comprehensive.

**Weaknesses:**

1. The paper lacks a thorough introduction to related work on the interpretation of scientific images, such as [1, 2, 3]. Furthermore, the distinctions between this work and ChartMimic remain unclear. To enhance clarity, Table 8 in the appendix should be included in the main body to better illustrate how this study compares to previous research.

2. The dataset primarily consists of query templates, which limits its generalizability for evaluating other models due to the heavy reliance on human (scientist) assessments.

3. Given the focus on scientific images, it is crucial to evaluate the model’s understanding of domain-specific scientific knowledge embedded within these images, rather than solely assessing spatial, numerical, or attribute-based features (e.g., color, size, shape). Images from domains such as biology, chemistry, physics, and materials science contain nuanced, domain-specific information that should be considered to provide a more complete assessment.

[1] Roberts, Jonathan, et al. "SciFIBench: Benchmarking Large Multimodal Models for Scientific Figure Interpretation." arXiv preprint arXiv:2405.08807 (2024).
[2] Li, Zekun, et al. "Mmsci: A multimodal multi-discipline dataset for phd-level scientific comprehension." arXiv preprint arXiv:2407.04903 (2024).
[3] Wang, Zirui, et al. "Charxiv: Charting gaps in realistic chart understanding in multimodal llms." arXiv preprint arXiv:2406.18521 (2024).

**Questions:**

1. How does this benchmark dataset differ from ChartMimic, and why should it be discussed extensively in the main body?
2. How can this benchmark be used to evaluate other models? Is evaluating only GPT-4 and LLaMA sufficient, or should more multimodal LLMs be tested?

---

> ### Author Response · Authors · 2024-11-22
> **Related work**
>
> > The paper lacks a thorough introduction to related work on the interpretation of scientific images, such as [1, 2, 3]. Furthermore, the distinctions between this work and ChartMimic remain unclear. To enhance clarity, Table 8 in the appendix should be included in the main body to better illustrate how this study compares to previous research.
>
> Thanks for pointing out these papers. We will ensure to include them in the final version. We would like to highlight the differences distinctly separating us from them: Images in three papers all serve as model input to test models image **understanding** ability, but we test models’ image **generation** performance (image as output). In NLP, there’s a well-accepted distinction between understanding (NLU) and generation (NLG).
>
> Paper [1] is on scientific figure interpretation with images and text (multi-choice question) as the input, while we focus on text-to-image generation.
>
> Paper [2] is also about understanding rather than generation. It uses multiple-choice questions to assess models’ understanding ability.
>
> Paper [3] is also not about generating images but chart understanding, a text+image to text task (image + question as input and requires models to provide textual answers).
>
> The paper ChartMimic is particularly about charts and it generates only code and only in English. It further does not evaluate particular understanding dimensions as we do.
>
> Thus, our benchmark differs decisively in testing whether models can generate scientific images, given input prompts with particular understanding dimensions. Generating is typically considered (a lot) harder than understanding.

---

> ### Author Response · Authors · 2024-11-22
> **Dataset**
>
> > The dataset primarily consists of query templates, which limits its generalizability for evaluating other models due to the heavy reliance on human (scientist) assessments.
>
> We would like to point out a confusion here between 1) evaluation and 2) benchmark construction. We leverage humans in order to ensure the reliability of the evaluation. However, in principle there is no limitation in using automatic metrics for our tasks, if the metrics perform well (our current results show that they don’t - but better metrics might perform better in the future). The benchmark generation, on the other hand, tests particular dimensions of understanding. We construct them automatically using templates; this is a common alternative to collecting data “in the wild”. Its advantage is that we can focus on test cases we’re interested in. This approach may be seen as a form of “checklist” type evaluation paradigm, popularized in NLP a while ago already: https://aclanthology.org/2020.acl-main.442/. We will ensure to add information on the above points in the final version to avoid confusion.
>
> If the reviewer meant to say that we lack reference images, then we have them at least for a subset of the data.

---

> ### Author Response · Authors · 2024-11-22
> **Domain-specific scientific knowledge**
>
> We want to clarify that our objects (2D shapes, 3D shapes, matrices, graphs, tables, etc.) are indeed very heavily inspired by the scientific domain, in particular the Datikz dataset, which is composed primarily of computer science papers (due to the nature of arxiv) but also other fields like physics and mathematics. Adding further domain specific knowledge (e.g., from biology, chemistry) would certainly be an interesting avenue for future work, making the benchmark even harder for the models. Even so, we showed that the models largely fail on the selected subset of science we considered and assessed this for our three understanding dimensions.

---

> ### Author Response · Authors · 2024-11-22
> **General feedback**
>
> We thank the reviewer for their very valuable comments and try to address them as good as we can. If they think that their concerns are adequately addressed, we'd be happy if they raised their scores and are very open to further discussion.
>
> We also appreciate that they found our analysis and benchmark comprehensive.

---

> > ### Author Response · Authors · 2024-11-26
> > **Hope we've addressed your concerns**
> >
> > Dear Reviewer:
> > Thanks for taking your time to review our paper. If we have addressed your concerns, we would greatly appreciate your updated scores - or we'd be happy to address any further questions you may have.

---

### Official Review · Reviewer_hHH2 · 2024-11-05

**Soundness:** 2
**Presentation:** 3
**Contribution:** 2
**Rating:** 5
**Confidence:** 4

**Summary:**

This paper introduced a benchmark, ScImage, for evaluating models for generating scientific images from textual descriptions. This paper thoroughly investigated the existing datasets and benchmarks for scientific image generation, and pointed out there is no systematic evaluations for scientific text-to-image models. This paper evaluated 5 state-of-art models with 2 generation modes: text-image and text-code-image, while text-code-image includes TikZ and Python as coding language. The benchmark emphasizes comprehension along three dimensions: numeric, spatial and attribute. It contains a 404 hundreds queries in 101 templates. Human evaluation is based on correctiveness, relevance and scientific style. It is good including agreements analysis of the human annotations to make sure human evaluation is reliable. The 5 models are compared over these criteria, and compilation failure is pointed out as a drawback of text-code-image models. Detailed analysis is made on types of understanding, object categories, multilingual prompts, and different programming languages. A qualitative analysis is conducted and highlighted some typical failures about understanding physical world. The findings underscore the further exploration on generating scientific images with LLMs. The paper summarized some limitations, such as not utilizing chain-of-thought, not developing automatic metrics and potential ethic issues.
Overall, the paper is well written, and has the potential to contributes a new benchmark for evaluating models capability to generate scientific image, which enables more comprehensive analysis towards models, especially LLMs.

**Strengths:**

1. Originality: The paper presents a potentially valuable contribution by introducing a new benchmark, ScImage, specifically focused on evaluating scientific image generation from text descriptions. While benchmarks for general text-to-image generation exist, few are tailored for scientific image generation, where precision in spatial, numeric, and attribute comprehension is critical. This benchmark addresses a gap in the current evaluation landscape, particularly for multimodal LLMs.

2. Quality: The paper presents a clear structure and coherent organization. While it contains several commendable elements, there are areas where improvement is needed to enhance the overall quality of the research. Addressing these aspects could significantly strengthen the findings and their implications.

3. Clarity: The paper is clearly structured, with each section flowing logically into the next. The figures and examples enhance comprehension by illustrating key points effectively.

4. Significance: The benchmark’s potential to foster progress in scientific image generation could be significant, as this area has notable applications in research, education, and communication. By evaluating models on the specific demands of scientific image generation, such as spatial accuracy and numeric precision, this benchmark could aid in developing more capable multimodal models tailored to scientific needs.

**Weaknesses:**

There are some problems, which must be solved before it is considered for publication. If the following problems are well-addressed, I believe that the essential contribution of this paper are important for evaluating LLMs' capability of text to image generation.
1. Your manuscript needs careful editing and particular attention to spelling, for example, in Figure 7, "Llame" should be "Llama".
2. Concerns with Dataset Construction:  I question the methodology used in constructing the dataset. The dataset appears to be generated from templates, producing a variety of prompts but lacking ground truth. Additionally, the dataset's size is relatively small, with only around 400 examples, which contrasts starkly with related datasets mentioned in the paper that typically consist of tens of thousands of examples. This raises concerns about the quality of the generated prompts and whether the dataset is sufficient to adequately evaluate the model. I suggest to add a quality analysis to the dataset. For example, evaluating a subset of the dataset with domain experts to make sure the generates prompts make sense, and performing some analysis to ensure the diversity and representativeness despite of the small size of dataset.
3. Concerns with Evaluation Approach:  I am also skeptical about the evaluation methodology, as it relies solely on human assessments. The authors justify this choice by referencing limitations of automated evaluation metrics, particularly in the context of scientific figures where metrics like CLIP Score and FID Score may fall short. For example, they mention issues noted by the MMVP benchmark with automated evaluations, especially in accurately identifying specific directions like "up" and "down." However, citing one bad case is insufficient to conclude that automated evaluation is unreliable. I would suggest to provide more detailed and comprehensive reasoning for the necessity for human evaluation over automated metrics in this specific context.
4. Lack of Depth in Result Analysis:  The results section primarily describes the scores obtained by each model but lacks a deeper analysis of these results. I would suggest to add the possible explanation to the observed phenomenon, the underlying problem of the models, and how the results indicates the directions for improvement. That could offer valuable insights for future research, enhancing the paper’s overall impact and utility.

**Questions:**

Data Construction:
1. Can you provide a detailed explanation of the rationale behind constructing the dataset solely from generated prompts based on templates? What considerations were made regarding the lack of ground truth in this dataset? Additionally, have you considered using real-world scientific data to construct the prompts, which would provide a more robust basis for evaluation?
2. Given that the dataset is relatively small, with only around 400 entries, how do you justify its representativeness and reliability compared to other related datasets that typically contain tens of thousands of examples?
3. You mentioned that you are only using the most commonly encountered types listed in the taxonomy. How did you decide which ones are the most commonly encountered?

Human Evaluation:
You mentioned that automated evaluations can be unreliable, particularly in recognizing precise directions in text and images. Can you elaborate on the specific limitations of current automated metrics, especially in the context of evaluating scientific graphs? Why do you believe that a single bad case is sufficient to justify the exclusive reliance on human evaluation?

Evaluation Guideline:
I suggest revising the evaluation guidelines to recommend using the same prompt for generating images with different scores, rather than using different prompts for different scores. How would you justify this approach, and do you believe it could lead to a more reliable assessment of the generated images?

Result Analysis:
In the results section, there is a description of the experimental data but a lack of in-depth analysis of the results. Could you clarify the reasoning behind the evaluation results? How do you believe that this level of analysis contributes to the significance of your findings and their applicability to future research in this area?

---

> ### Author Response · Authors · 2024-11-22
> **Concerns with Dataset construction**
>
> Comparing to common tasks like sentiment analysis, 400 may sound like a small number but we would like to emphasize the challenges of creating and evaluating scientific tasks, which reflect the value of our work, and that the data set that we evaluated is actually much larger:
>
> 1) creating and evaluating a scientific text-to-image dataset requires expertise: We tax the value of our evaluation at roughly 3,000 USD, with up to 11 annotators (STEM PhD or faculty in our case) involved for up to 7 hours each, all working for a conservative estimate of 40 USD per hour (including taxes), on average. We added these numbers to the revised version. Note that scientific tasks require expert annotators as shown in Belouadi et al. (2024a,b) as otherwise the agreements are too low.
>
> 2) reasonable numbers compared to similar scientific domain tasks: while we have only 404 prompts, we provide human ground truth of evaluation for ~3k generated images in three dimensions in our dataset (as there are several models that generate for each prompt), all evaluated by up to 11 expert annotators. This size makes it comparable to similar datasets in the community, e.g. MMVP (Tong et al., 2024) with 300 image+question pairs, CONTEXTUAL (Wadhawan et al., 2024) with 506 samples, SciBench (Wang et al, 2024) with 869 cases. So, these numbers are actually smaller than ours (others may contain reference images - but so do we for a subset of highly scored images).
>
> 3) regarding “evaluating a subset of the dataset with domain experts”: we fully agree and want to point out that our prompt dataset has been fully checked (and even constructed) by computer science PhD and faculty. Diversity and representativeness is ensured by the data construction: e.g., we focus on the three dimensions of numeric, spatial and attribute understanding and their combinations and these dimensions are adequately represented (in roughly equal proportion), by construction. Of course, one might argue to include further dimensions - and we are open to this - which could yield additional insights into other model failures, but we think that the **current dimensions already make a strong point regarding limitations of text-to-image generation models in the scientific domain**. We also want to highlight that we provide statistics over different quantities of scientific graphs in the tables (e.g., sample sizes of different prompts relating to numeric understanding, etc.).
>
> 4) lacking ground truth: we admit that we lack ground truth for some of the images (but we have them indirectly for a large subset of images which are highly scored by at least one of the models). As a benchmark, what we contribute primarily is to provide Ground Truth Evaluation Scores from Experts for ~3k generated images which will facilitate the improvement of automated evaluation metrics. In addition to this, we will release highly scored reference images for a subset of the data.
>
> > Can you provide a detailed explanation of the rationale behind constructing the dataset solely from generated prompts based on templates? What considerations were made regarding the lack of ground truth in this dataset? Additionally, have you considered using real-world scientific data to construct the prompts, which would provide a more robust basis for evaluation?
>
> In dataset construction, our motivation was to provide clearly specified targeted dimensions of scientific image generation where we exactly know the attributes queried. This makes the evaluation specific & targeted. This contrasts with a “real-world” evaluation where potentially long query prompts are leveraged which may contain various dimensions at the same time and, thus, model failure may be a result of multiple aspects which cannot be clearly disentangled. For example, Datikz (Belouadi et al. 2024) provides a real-world dataset - while it inspired us in selecting scientific objects, it is unspecific and mixes different understanding properties in each prompt. Our evaluation framework is inspired by the famous CheckList methodology of Ribeiro (https://aclanthology.org/2020.acl-main.442/).

---

> ### Author Response · Authors · 2024-11-22
> **Concerns with Evaluation Approach**
>
> Thanks for pointing out the case of automatic evaluation metrics. We have now run 5 SOTA metrics on our data to further test the current automated metrics:  CLIPScore (vit-base-patch16 and vit-large-patch14) (Hessel et al., 2021), ALIGNScore (Saxon et al., 2024) and PickScore (Kirstain et al., 2023), as well as the multimodel LLM based metric DSG (Cho et al.,2024) with Gemma-2-9B-SimPo (Team, 2024; Meng et al., 2024). They all fail, with the correlation below 0.27 with humans. This is strong evidence that current metrics aren’t adequate for scientific image generation evaluation and that we need humans. In future work, we would like to provide such metrics with good correlation to human assessments, but believe that this is beyond the scope of our current research.
>
> >  Can you elaborate on the specific limitations of current automated metrics, especially in the context of evaluating scientific graphs? Why do you believe that a single bad case is sufficient to justify the exclusive reliance on human evaluation?
>
> In general, automatic metrics have low correlations with humans in many domains, even in very heavily researched machine translation (see WMT shared tasks where metrics often have correlations of at most 0.3-0.4, depending on the evaluation scenario). While we had not verified this empirically for our case, we have done so in the revision. As to why they fail, for our use case one important reason is the lack of exposure to scientific data, we believe.
>
> > Evaluation Guideline: I suggest revising the evaluation guidelines to recommend using the same prompt for generating images with different scores, rather than using different prompts for different scores. How would you justify this approach, and do you believe it could lead to a more reliable assessment of the generated images?
>
> We do not understand what you mean but speculate that there might be a gross misunderstanding here. To clarify, we prompt the models and evaluate the output by human experts (and now also automatic metrics). Humans assign scores based on the different evaluation dimensions. We do not prompt scores. We're happy to give an updated answer if you could clarify the question.

---

> ### Author Response · Authors · 2024-11-22
> **Lack of Depth in Result Analysis**
>
> We want to clarify that we have conducted multiple analyses in the paper. For example, we analyze the impact of object types (Table 5 in the revision), the impact of the input language (Table 6), the impact of understanding dimensions (Table 4 and Figure 3), differences in output code (Table 7) besides a quantitative analysis. For instance, our findings indicate that spatial understanding is the most challenging understanding dimension for text-code-image models while numerical understanding is most challenging to text-image models (direct generation of images). In addition, we have now conducted a further analysis which finds the availability of Python/TikZ libraries contributes to some failures. For instance, the absence of a Python library specifically for drawing matrices results in GPT-4’s Python code performing worse than TikZ code.
>
> Our analysis contributes to better understanding which output format is better suited for which understanding dimensions, differences in input source languages and differences between scientific objects generation. We think our analysis section is comparatively long given the page limit of conferences like ICLR.

---

> ### Author Response · Authors · 2024-11-22
> **General feedback**
>
> We want to thank the reviewer for outlining several aspects which deserved further clarification. We're happy to say that we have run automatic metrics, report their results, and clarify the dataset construction. We feel that many concerns of the reviewer may be addressed now, and we would hope that s/he would increase their score if they think their concerns are addressed.
>
> Additionally: Thank you very much for acknowledging our “valuable contribution” on the “new benchmarking”. We are encouraged that our paper is found clearly written with notable applications. Thank you for pointing out the typo (even in the appendix!), we will ensure it will be addressed in the final version and we would like to address the reviewer’s remaining concerns in individual comments.

---

> ### Comment · Reviewer_hHH2 · 2024-11-26
>
> Thank you for your reply! The explanation regarding data construction and the evaluation approach has largely addressed my concerns. I believe it would strengthen the justification for using human evaluation to explicitly mention that the automatic metrics failed.
>
> By evaluation guideline I mean Table 13: Correctness Guideline Examples. I see the prompts for examples of different scores are different, I think it would be better to use the same prompt and see the failure cases at different scores.

---

> ### Author Response · Authors · 2024-11-26
>
> Thanks a lot for the feedback and adjusting your score.
>
> > . I believe it would strengthen the justification for using human evaluation to explicitly mention that the automatic metrics failed.
>
> Thanks a lot - we added this on l.318ff in the revision.
>
> > By evaluation guideline I mean Table 13: Correctness Guideline Examples. I see the prompts for examples of different scores are different, I think it would be better to use the same prompt and see the failure cases at different scores.
>
> We see, thanks a lot. That does make sense and might give better guidelines. However, please note that our agreements are generally quite good (see Table 2); in the revised version, we also report chance-adjusted interrater reliability scores. Our agreements for correctness are ~0.6 weighted kappa and above 0.7 for Spearman/Pearson.
>
> So, we do believe that our annotators are quite consistently annotating and thus we didn't refine our guidelines further. However, in retrospect, your suggestion might have been better, leading potentially to slightly higher agreements.

---

### Meta-Review · Area_Chair_uAmi · 2024-12-23

**Metareview:**

The works presents a new benchmark to evaluate text-to-image capabilities for scientific images (2D/3D shapes, charts, graphs, matrices, tables, etc). The paper tested various models like GPT4, Llama, Stable Diffusion. The paper evaluates an important aspect of image generation (generating scientific images instead of natural images). Sharing this benchmark with has the potential to engage the community in a new interesting problem, and for those reasons I recommend accepting the paper.

One of the main concerns raised by reviewers is the need for manual human evaluation and it's high cost. It's unclear ow much will this scale when the community wants to evaluate the methods. Current automatic metrics show poor correlation with humans, which make this a challenging problem. Nevertheless, reviewers agreed that the contributions are significant to merit an acceptance.

**Additional Comments On Reviewer Discussion:**

Templates for the dataset: Reviewers raised concerned that the benchmark contains templates which limits generalizability. The authors argued that using templates allow to control the diversity and difficulty of the dataset, testing for key abilities which might be difficult to control for images in the wild.

Other domains: Reviewer 4MC8 raised concerns that the benchmark should contain domain-specific nuances (especially from fields like biology, chemistry). The authors argue that while dataset is primarily composed of computer science papers from arxiv, there are some instances which are from physics and mathematics. Additionally, the models largely fail on the instances in the benchmark, so even the current benchmark is challenging and the difficulty can be increased as future work.

Reviewer prga asked several clarifiying questions about addtitional baselines, metrics, automatic evaluations, and performance of open-weights model. The model clarified those by adding extra material in the appendix. As requested by the reviewer, the authors should perform all promised updates for the camera ready, and maybe additionally run an experiment with state-of-the-art open-weights model like Qwen-2.5 or similar model.

---

### Decision · Program_Chairs · 2025-01-22

Accept (Poster)